# *Chia* Seed (*Salvia hispanica*) Attenuates Chemically Induced Lung Carcinomas in Rats through Suppression of Proliferation and Angiogenesis

**DOI:** 10.3390/ph17091129

**Published:** 2024-08-27

**Authors:** Naglaa A. Ali, Ghada H. Elsayed, Safaa H. Mohamed, Asmaa S. Abd Elkarim, Mohamed S. Aly, Abdelbaset M. Elgamal, Wael M. Elsayed, Samah A. El-Newary

**Affiliations:** 1Hormones Department, National Research Centre, El-Bouhoths St., Dokki, Giza 12622, Egypt; almardeyah@gmail.com (N.A.A.); ghadanrc@yahoo.com (G.H.E.); rise_sun_1982@hotmail.com (S.H.M.); 2Stem Cells Lab, Centre of Excellence for Advanced Sciences, National Research Centre, Dokki, Giza 12622, Egypt; 3Chemistry of Tanning Materials and Leather Technology Department, National Research Centre, Giza 12622, Egypt; asmaa_nrc@yahoo.com; 4Department of Animal Reproduction and Artificial Insemination, National Research Centre, 33 El-Bohouth St., Dokki, Giza 12622, Egypt; mohamedaly_nrc@yahoo.com; 5Department of Chemistry of Microbial and Natural Products, National Research Centre, 33 El-Bohouth St., Dokki, Giza 12622, Egypt; algamalgene@yahoo.com; 6Chemistry of Medicinal Plants Department, National Research Centre, Giza 12622, Egypt; waelmatar@hotmail.com; 7Medicinal and Aromatic Plants Research Department, National Research Centre, El-Bouhoths St., Dokki, Giza 12622, Egypt

**Keywords:** lung cancer, chia seed (*Salvia hispanica*), cytotoxicity, 4-(Methylnitrosamino)-1-(3-pyridyl)-1-butanone (NNK), LC-Mass

## Abstract

In 2022, 2.5 million cases of lung cancer were diagnosed, resulting in 1.8 million deaths. These statistics have motivated us to introduce a new natural product which is feasible in lung cancer therapies. This comprehensive study was performed to study the effects of chia seed extracts (70% ethanol and petroleum ether) on lung cancer in vitro and in vivo models. The invitro cytotoxicity activity of the chia extracts was studied in lung cancer cell lines (A549 cells). After 48 h, chia alcohol and ether extracts showed more inhibitory influence (IC_50_, 16.08, and 14.8 µg/mL, respectively) on A549 cells compared to Dox (IC_50_, 13.6 µg/mL). In vivo, administration of chia alcohol and ether extracts (500 mg/kg/day, orally for 20 weeks) recovered 4-(Methylnitrosamino)-1-(3-pyridyl)-1-butanone (NNK)-induced lung cancer, as a significant reduction in the lung cancer biomarkers, including the relative weight of the lung (20.0 and 13.33%), ICAM(31.73 and 15.66%), and c-MYC (80 and 96%) and MMP9(60 and 69%) expression genes, and improvement in these changes were observed by histopathological examinations of the lung tissues compared to the lung control. Chia seeds fought lung cancer via suppression of proliferation, angiogenesis, inflammation, and activation apoptosis. These activities may be attributed to the chemical composition of chia, which is identified by LC-Mass, such as caffeic acid, vanillic acid, kaempferol-3-*O*-glucuronide, and taxifolin. Finally, we can conclude that chia seeds have an anti-lung cancer effect with a good safety margin.

## 1. Introduction

A malignant tumor that starts in the lung is called lung cancer. Genetic damage to the DNA of lung cells is the primary cause of lung malignancies, and this damage is frequently made worse by smoking cigarettes or breathing in chemicals that can cause cancer. Tumors destroy lung function when they spread across the lung without therapy. Lung cancers eventually spread to other sections of the body through metastasis. Based on the cells that give rise to them, lung tumors are categorized. Squamous-cell carcinomas, large-cell carcinomas, and adenocarcinomas make up the remaining 85% of lung cancers, which are not small-cell lung cancers. Small-cell lung cancers account for around 15% of lung cancers [1]. Most cases of early lung cancer are symptomless. Most patients who have cancer eventually have nonspecific respiratory symptoms, such as coughing, shortness of breath, and chest pain. Depending on the tumor’s size and location, a wide range of symptoms may accompany it. Many experience symptoms as a result of metastases, which typically affect the adrenal glands, brain, bones, and liver. Lung cancer is frequently fatal even with treatment; only around 19% of patients survive five years following diagnosis [2].

Lung cancer was the most frequently diagnosed cancer in 2022, responsible for almost 2.5 million new cases, or one in eight cancers worldwide (12.4% of all cancers globally), followed by cancers of the female breast (11.6%). Lung cancer was also the leading cause of cancer death, with an estimated 1.8 million deaths (18.7%), followed by colorectal (9.3%) and liver (7.8%) cancers. Lung cancer was the most frequent cancer in men. It is uncommon for those under 40 to be diagnosed with lung cancer; the average age of death is 72, and the typical diagnostic age is 70. The five-year survival rate is between 10% and 20% in the majority of nations. However, incidence and results differ significantly by location based on tobacco usage patterns. For men, it ranks as the most common cancer, while for women, it ranks fourth. In both sexes, it accounts for 13.0% of all cancer cases. In poorer nations, lung cancer cases account for 55% of cases [3,4].

In Egypt, lung cancer is one of the most common cancers, accounting for 5.0–7.0% of all cancers. Its incidence increased during 1980–2014, from 11.9 to 63.3/100.000 population for men and from 3.7 to 13.8/100.000 population for women. Lung cancer ranks fifth in males and both sexes and ninth among females. Also, lung cancer is a leading cause of death (25.0% of all cancer deaths) [5].

One of the main nitrosamines peculiar to tobacco that is formed from nicotine is called nicotine-derived nitrosamine ketone (NNK). It is crucial to the formation of cancer. The pyrrolidine ring must open for nicotine to be converted to NNK. Being a mutagen, NNK is known to induce polymorphisms in the human genome. Research revealed that gene polymorphisms in cells caused by NNK have a role in cell division, growth, and proliferation. There are several pathways involving cell proliferation that are dependent on NNK. When it comes to gene silencing, alteration, and functional disruption that leads to carcinogenesis, NNK is a crucial player [6].

Because they have fewer adverse effects and are more widely accepted, herbal plants have drawn a lot of interest recently. Repurposing medications that are currently available and using medicinal plants have emerged as viable approaches to discovering novel therapeutic possibilities [7]. Several medicinal plants exhibited anticancer activities, including anti-prostate cancer [8], anti-breast cancer [9], anti-colon cancer [10], and anticancer apoptosis [11].

Chia seeds are the item under test in this investigation. Native to northern Guatemala and southern Mexico is the chia plant (*Salvia hispanica*). An annual herbaceous plant in the *Lamiaceae* family is called *Salvia hispanica* [12]. Chia seeds are presently utilized for their nutritional and therapeutic qualities, such as their ability to promote intestinal regularity, suppress hunger, aid in weight loss, increase athletic endurance, and lower blood sugar levels [13]. Chia seeds are considered to have a high nutritional value because of their high fat and dietary fiber content. Fascinating is the fatty acid profile, such as α-linolenic acid (ALA), which makes up almost 60% of all fatty acids. Chia seeds are also a good source of vitamins (B1, B2, and niacin), minerals (phosphorus, calcium, potassium, and magnesium), and plant protein. Additionally, chia seeds are a rich source of particularly intriguing groups of phytocompounds known as isoflavones (daidzein, glycitein, genistein, and genistin) and polyphenols (gallic, caffeic, chlorogenic, cinnamic, ferulic acids, quercetin, kaempferol, epicatechin, rutin, apigenin, and p-coumaric acid). In addition to tocopherols, seeds contain sterols such as stigmasterol, β-sitosterol, and D5-avenasterol. Salvia plants are used to treat a variety of pathological disorders, including cancer, brain dysfunction, and atherosclerosis. They also have potent antioxidant and antibacterial properties [13]. During both short-term (six weeks) and long-term (12 weeks) dietary regimens, chia seeds show hypoglycemic and hypolipidemic effects [14,15]. Acetylcholine-induced enhanced aortic relaxation and higher NO release were seen in rabbits fed chia seed oil [13]. Chia oil has immunomodulatory properties [16]. Despite the high nutritional value of chia seeds, few studies have addressed their therapeutic functions. From this standpoint, the current study was conducted.

This study aimed to (i) evaluate the anticancer effect of chia seeds ethanolic and petroleum ether extracts in NNK-induced lung cancer in rats, (ii) study the possible mechanisms of the extracts in lung cancer, and (iii) identify the chemical profile of the extract using HPLC/QTOF-HR-MS/MS analysis.

## 2. Results

Several studies have shown the anticancer effect of chia oil in lung cancer, but on an in vitro scale using A549 cell lines. This study is the first to examine the anticancer effect of chia seed extracts, petroleum ether, and 70% ethanol on lung cancer on an in vivo scale. It is also the first study to use the following model: 4-(Methylnitrosamino)-1-(3-pyridyl)-1-butanone (NNK)-induced lung cancer in rats. In addition, we studied how chia extracts work in treating lung cancer. Our journey began with determining the activity of these extracts in vitro, using the lung cancer cell line (A549 cells) as our testing ground. Once we successfully demonstrated the potent activity of these extracts in vitro, we incorporated them into an in vivo study using NNK-induced lung cancer rat models.

To evaluate the anti-lung cancer effect, we estimated three anti-lung cancer biomarkers: Intercellular Adhesion Molecule-1(ICAM-1), MYC proto-oncogene, bHLH transcription factor(c-MYC), and Matrix metalloproteinase-9 (MMP9) genes. Also, we studied the anti-lung cancer mechanism of chia seed hydroethanolic extract and petroleum ether extracts by examining the effect of chia seed materials on (i) inflammation (Interleukin 6 and Interleukin 1 beta (IL-6 and IL-1β, the pro-inflammation biomarkers), (ii) angiogenesis (vascular epithelial growth factor, VEGF, the angiogenic biomarker), (iii) apoptosis (B-cell leukemia/lymphoma two protein (BCL2), the anti-apoptotic biomarker), (iv) proliferation (Antigen Kiel 67 (Ki67) biomarker), and (v) oxidative stress. Also, we estimated the effect of chia seed materials on the safety profile of rats to know their toxicity or safety. We mention the relationship between biomarkers and lung cancer in the discussion section. Finally, we studied the chemical composition of alcoholic and petroleum ether extracts by LC-QTOF-HR-MS/MS.

### 2.1. In Vitro Study

#### Cytotoxicity Activity of the Chia Extracts against Lung Cancer Cell Line (A549 Cells)

The A549 cell line was administered by the tested extracts at varying concentrations ranging from 25 to 200 µg/mL. Doxorubicin (Dox) was used as a reference drug, and it significantly inhibited the growth of A549 cells at 48 h in comparison with control values. After 48 h of treatment, the use of DMSO as a solvent had an insignificant effect on the viability of A549 cells. All of the investigated extracts significantly altered the suppression of cell growth when compared to the control cells (Figure 1). In comparison to Dox (IC_50_ = 13.6 µg/mL), chia alcohol and chia ether (IC_50_ = 14.8, 16.08 µg/mL, respectively) exhibited the most potent cytotoxic activity against A549 cells after 48 h.

### 2.2. In Vivo Study on NNK-Induced Lung Cancer in Rat Model

#### 2.2.1. The Effect of Chia Extract on Lung Cancer Biomarkers

Lung cancer induction significantly elevated the relative weight of the lung (g/100 g body weight) from 0.88% ± 0.012 in the negative control to 1.05% ± 0.014 in lung cancer control rats (*p* ≤ 0.05) (Figure 2A). Administration of chia materials: ether or alcohol extracts significantly reduced the relative weight of the lung (0.84% ± 0.04 and 0.91% ± 0.02) compared to the cancer control. Compared to the negative control, chia materials did not record a significant effect on the relative weight of the lungs of positive groups.

ICAM-1, the lung cancer biomarker, was significantly increased by NNK injection. The ICAM-1 level of the cancer control rats (582.95 ± 7.17 pg/mL) was greater than that of the negative control (361.55 ± 2.19 pg/mL) by about 37.98% (*p* ≤ 0.05) (Figure 2B). Chia treatment caused a recovery effect as there was a significant decrease in ICAM-1 level in the chia ether group (397.97 ± 7.55 pg/mL) and the chia alcohol group (491.67 ± 6.54 pg/mL) in the comparison to the lung cancer control. The ICAM-1 of chia ether positive control significantly decreased compared to the negative control. Meanwhile, the chia alcohol positive control decreased insignificantly compared to the negative control.

Lung cancer induction dramatically activated c-MYC and MMP9 gene expression by about 161 and 64% compared to those of the negative control group. Our results showed that chia extracts; alcohol and ether significantly reduced expression levels of the studied genes c-MYC (by about 96 and 80%) and MMP9 (by about 69 and 60%) compared to the lung cancer group (*p* ≤ 0.05) (Figure 2C,D). On the other hand, as compared to the negative control group, chia alcohol, and ether extracts revealed a nonsignificant decrease in expression levels of the c-MYC gene (Figure 2C), while chia alcohol extract showed a significant reduction in expression levels of the MMP9 gene (*p* ≤ 0.05) (Figure 2D).

#### 2.2.2. The Effect of Chia Extract on Proliferation

Lung cancer induction encouraged proliferation, as represented by the significant increase in Ki67, the proliferation biomarker (Figure 3A). The Ki67 levels were remarkably minimized in groups treated with chia ether (5.85 ± 0.4 ng/mL) and chia alcohol (5.36 ± 0.3 ng/mL) compared to the Ki67 level of the lung cancer group (8.28 ± 0.49 ng/mL). The Ki67 level of chia ether and chia alcohol positive controls decreased significantly compared to the negative control.

#### 2.2.3. The Effect of Chia Extract on Apoptosis

Lung cancer induction considerably activated BCl2 gene expression by about 40% compared to the negative control (*p* ≤ 0.05) (Figure 3B). Chia alcohol and ether extracts revealed a pro-apoptotic effect via a significant reduction in expression levels of the BCL_2_ apoptotic gene by about 52 and 56% for chia ether and chia alcohol extracts compared to the control and lung cancer groups (*p* ≤ 0.05) (Figure 3B).

#### 2.2.4. The Effect of Chia Extract on Angiogenesis

Compared to the VEGF level in the negative control group, lung cancer induction significantly increased VEGF by about 27.65%. Chia extract recorded an anti-angiogenic effect, resulting in a significant reduction in VEGF, the angiogenic biomarker (Figure 3C). The VEGF level significantly decreased by chia alcohol (181.44 ± 3.16 pg/mL) and ether (186.64 ± 4.35 pg/mL) extract relative to the cancer control: 233.26 ± 11.75 pg/mL (*p* ≤ 0.05). The VEGF of the alcohol extract positive group was significantly elevated (192.85 ± 5.21 pg/mL); meanwhile, ether extract was insignificantly elevated (177.79 ± 7.78 pg/mL) compared to the negative control.

#### 2.2.5. The Effect of Chia Extract on Inflammation

Exposure to NNK significantly affected inflammation as a significant increase in IL-6 and IL-1β, compared to the negative control. Chia alcohol and ether extracts exhibited anti-inflammatory effects that significantly reduced IL-6 and IL-1β levels (Figure 3D,E). Chia ether significantly minified IL-6 (75.20 ± 0.96 ng/mL) and IL-1β (59.98 ± 2.16 pg/mL) levels more than chia alcohol (80.73 ± 0.73 ng/mL and 61.35 ± 1.63 pg/mL, respectively), compared to the cancer lung control, 67.10 ± 1.72 ng/mL and 69.08 ± 0.83 pg/mL, respectively.

The anti-inflammatory effect of chia alcohol on IL-1β was more effective than IL-6. The IL-1β of chia alcoholic (61.35 ± 1.63 pg/mL) and ether (59.98 ± 2.16 pg/mL) extracts was returned to the normal range and significantly minified lower than the negative control: 66.93 ± 1.87 pg/mL (*p* ≤ 0.05). In the positive groups of chia alcohol and ether extracts, IL-6 was considerably elevated (70.24 ± 1.20 and 71.41 ± 1.39 ng/mL, respectively); meanwhile, IL-1β significantly lessened (57.16 ± 1.58 and 57.11 ± 0.78 pg/mL, respectively), compared to the negative control: 67.10 ± 1.72 ng/mL and 66.93 ± 1.87 pg/mL, respectively.

#### 2.2.6. The Effect of Chia Extract on Antioxidants

Lung cancer induction significantly depleted the concentration of glutathione (GSH) and significantly inhibited the activities of its related enzymes, glutathione reductase (GR), glutathione S transferase (GST), and glutathione peroxidase (GPx), in lung cancer, compared to the negative control group (*p* ≤ 0.05) (Table 1). Chia extracts showed antioxidant capacities as a significant increase in GSH concentration and GR, GST, and GPx activities compared to lung cancer control. The chia extract returned GSH and related enzymes close to normal ranges, particularly the chia ether extract. In addition, chia extracts significantly elevated GSH and related enzymes more than those of the negative control.

#### 2.2.7. The Effect of Chia Extract on the Safety Profile of Treated Rats

Administration of chia seed ether or alcohol extracts for five months (500 mg/kg/day) did not show any toxic effect on the liver or kidney. The liver and kidney performance of rats treated with chia in the positive controls was like those of the negative control. No significant differences were noticed between serum total protein and its fractions, albumin and globulin, uric acid, urea, and creatinine concentrations, and AST and ALT activities of chia ether or alcohol extracts and the negative control (Table 2).

Interestingly, chia materials, either ether extract or alcohol extract, improved the liver and kidney performance of NNK-induced lung cancer rats treated with chia. NNK injection caused toxicity in the liver and kidney, where liver and kidney functions were disrupted, compared to the negative control. NNK injection significantly decreased serum total protein and its fractions, albumin, and globulin in lung cancer control rats by about 25.70, 24.85, and 26.34%, respectively, compared to the negative control (*p* ≤ 0.05). Also, serum AST and ALT significantly elevated as a response to NNK injection by about 31.69 and 21.20%, respectively, compared to the negative control values (*p* ≤ 0.05). In lung cancer control rats, renal functions were disrupted by a significant rise in uric acid and urea levels compared to the corresponding value in the negative control (26.37 and 14.80, respectively).

On the contrary, chia extracts, either ether extract or alcohol extract, recorded a considerable improvement in liver and renal functions in treated rats. Chia extracts exhibited a high safety margin and returned liver and renal performance towards normal ranges in comparison to the negative control (Table 3).

#### 2.2.8. Histopathological Study

Gross pathological observations showed the presence of multiple small and large lung nodules. The histopathological examination data suggests that exposure to NNK for 20 weeks in rats could induce lung tumors and various types of lung cancer and carcinoma in a time-dependent manner. Histopathological examination of lung sections of the negative control and both chia-treated groups showed unremarkable histopathological alterations with typical alveolar architecture and thin interalveolar septa (Figure 4A (**1**–**6 respectively**)). However, examination of lung tissue sections of the lung cancer control group that received successive toxic doses of NNK showed moderate to severe hyperplasia, adenoma, and adenocarcinoma, where most of the bronchioles were degenerated and necrotic with massive interstitial macrophage infiltration within the alveolar spaces, and most of the alveolar structures were absent and replaced by proliferated cells showing cellular atypia, squamous-cell metaplasia, and squamous dysplasia. Some examined sections showed alveolar macrophages as well as inflammatory cell infiltration, alveolar hemorrhage, emphysema, squamous-cell carcinoma, and adenocarcinoma (Figure 4A (**7**, **8**)).

Conversely, examined lung tissue sections of the chia ethanolic extract-treated group that received successive toxic doses of NNK and were treated with chia hydroalcoholic extract showed a reduced alveolar dysplasia, and improved lung structures associated with mild hyperplasia in the epithelial cell lining, mildly congested blood vessels, and mild macrophage infiltration (Figure 4A (**9**, **10**)). Treatment with chia ether extract moderately ameliorated the harmful effects of NNK on the lungs of rats, where examined sections showed mild bronchiolar degeneration and necrosis with mild interstitial macrophage infiltration (Figure 4A (**11**, **12**)).

The lung cancer control group received successive toxic doses of NNK, showing moderate to severe hyperplasia, adenoma, and adenocarcinoma with massive interstitial macrophage infiltration within the alveolar spaces, and with proliferated cells showing cellular atypia, squamous-cell metaplasia, and squamous dysplasia. Some examined sections showed alveolar hemorrhage and emphysema (**7**, **8**). The chia ethanolic extract-treated group showed reduced alveolar dysplasia and improved lung structures associated with mild hyperplasia in the epithelial cell lining, mildly congested blood vessels, and mild macrophage infiltration (**9, 10**). The chia ether extract-treated group moderately alleviated the harmful effects of NNK on the lungs of rats, where examined sections showed mild bronchiolar degeneration and necrosis with mild interstitial macrophage infiltration (**11**, **12**). (1,3,5,7,9,11 X 100 and 2,4,6,8,10,12 X 200).

The lung cancer control group received successive toxic doses of NNK, showing the disordered systemic arrangement of the stages of spermatogenesis, disorganized seminiferous tubules, severe vacuolar degeneration, and necrosis of spermatogonial cells, as well as desquamation of spermatogonial cells lining seminiferous tubules with subcapsular hemorrhage and marked intertubular edema (**7**, **8**). Chia ethanolic extract and chia ether extract-treated groups showed marked restoration of the normal histologic architecture of the testis (**9**, **10**, **11**, **12**, **respectively**) (1, 3, 5, 7, 9, 11 X 100 and 2, 4, 6, 8, 10, 12 X 200).

### 2.3. Chemical Profiling of Significant Anticancer Lung Polyphenolics from Ethanol Extract of Egyptian Salvia hispanica Seeds

Herbal medicines and derivative products are widely employed as therapeutics in many countries. Their global use has expanded during the last decade. These herbs contain hundreds of active components that are almost impossible to detect. The chromatographic fingerprint technique has recently been recognized as an effective tool for the systemic characterization and identification of plant extract constituents [17]. Various mass spectrometric techniques have gradually been used to investigate medicinal plants and profile their secondary metabolites. Chromatography has recently embraced high-performance liquid chromatography/quadrupole time-of-flight mass spectrometry (HPLC/QTOF-MS) technology [18]. The initial motivation for this work was to use *Salvia hispanica* (Chia) seeds belonging to the mint family (*Lamiaceae*) to identify a viable cure for a dangerous lung cancer disease. Therefore, our goal was to identify the most effective anticancer lung compounds.

Although researchers have identified a number of beneficial chemical substances in *Salvia hispanica* (chia) seeds, few studies have investigated the whole polyphenol profile. Therefore, our goal was to identify the most effective anticancer lung compounds in *Salvia hispanica* seeds using HPLC/QTOF-MS/MS with positive (+ve) and negative (−ve) dual ionization modes based on the chemical formula, ionization modes (−ve and +ve), retention times (Rt), and exact mass detected the ppm error (between actual mass and observed mass) for all phytoconstituents. The limit of detection for each peak of the compounds was established with the MS/MS fragment ions by matching published data with reference compounds’ spectra. The total ion current (TIC) and base peak MS-chromatograms (BPCs) in both −ve and +ve ionization modes (Figure 5 and Figure 6) showed that the Egyptian *Salvia hispanica* (Chia) seeds contain a high concentration of polyphenols, including phenolic acids, flavonoids, and their glycosides, as well as tannins, anthocyanins, stilbene, tetrahydrofuran lignin, coumarin, alkaloids, and diterpenes. They were grouped and categorized into various classes based on retention time (Rt) (Table 4).

The chemical structures of the various polyphenolics were determined by fragment pattern analysis, in which glycosides of flavonoids were cut out from their structures as hexose, rhamnose, pentose, glucuronic acid, and neohesperidoside (*m*/*z* 162, 146, 132, 176, and 308) [19].

In this work, all of the separated compounds were detected and tentatively identified for the first time in the ethanol extract of Egyptian *Salvia hispanica* (Chia) seeds. The phytoconstituents were characterized using the negative (−ve) ion mode after the chromatographic conditions were refined. This mode yielded more peaks than the positive (+ve) ion mode (Figure 6). Taking into account [M-H]-ions, mass errors for all compounds were less than ±10 milli-*m*/*z* units, reference standards, published fragmentation techniques, and documented data were used to characterize the extracted and matched compounds further.

As a result, a total of 120 peaks were detected and tentatively identified (Table 4). The metabolites were classified into multiple types of natural products, which included 21 acids, nine phenolic acid hexosides (mono and di), and 66 different flavonoid compounds, comprising 12 derivatives of kaempferol, 11 derivatives of quercetin, 3 derivatives of myricetin, 18 metabolites of flavones, 2 flavanones, 2 flavan-3-ols, 10 aglycones, 2 tannins, 2 anthocyanin glycosides, 2 chalcones, and 2 stilbenes. The other types included 2 coumarins, 3 alkaloids, 16 diterpenes, 1 amino acid, and two organic compounds.

### 2.4. Characterization of Highly Concentrated Metabolites in Egyptian Chia Salvia hispanica Seeds by HPLC/QTOF-MS/MS Analysis

#### 2.4.1. Phenolic Acids

Two subclasses of phenolic acids were tentatively identified in both −ve and +ve modes, namely hydroxybenzoic acid and hydroxycinnamic acids (Table 4).

Caffeic acid (**5**), ferulic acid (**6**), rosmarinic acid (**7**), coumaric acid (**8**), przewalskinic acid (**10**), danshensu (**11**), D-(+)-galacturonic acid (**15**), salvianolic acid A (**16**), salvianolic acid F (**17**), salvianolic acid L (**18**), isosalvianolic acid B (**19**), salvianolic acid D (**20**), lithospermic acid (**21**), ferulic acid *O*-hexoside (**24**), caffeic acid *O*-hexoside (**26**), caffeoylquinic acid (**28**), and caffeic acid di rhamnoside (**30**) were tentatively identified as hydroxycinnamic acid derivatives.

Protocatechueic acid (**9**), vanillic acid (**12**), P-hydroxybenzoic acid (**13**), protocatechuicaldehyde (**14**), protocatechueic acid-*C*-hexoside (**22**), vanillic acid-*O*-hexoside (**23**), protocatechueic acid-*O*-hexoside (**25**), and hydroxybenzoic acid hexoside (**27**) were all tentatively recognized as derivatives of hydroxybenzoic acid [20].

Phenolic acids are largely fragmented through the cleavage of CO_2_ and the hexosyl moiety from parent ions. The loss of CO_2_ (44 Da) from the carboxylic acid group is a prominent aspect of the fragmentation patterns produced by phenolics [21,22]. In compliance with this rule, compounds (**5**–**13**, Appendix A), which are highly concentrated free phenolic acids, were detected in both −ve and +ve modes. They displayed characteristic product ions at *m*/*z* 161.0420, 135.0402, 177.0700, 175.0609, 157.0389, 149.0555, 151.0749, 315.1075, 119.0504, 147.0315, 135.000, 137.0261, 109.0315, 313.1408, 152.9751, 112.9854, 181.1058, 149.0223, 122.9672, and 121.0388, identical to the equivalent loss of CO_2_ (−44 Da) and H_2_O (−18 Da) [23].

On the other side, compounds (**14**–**21**) were tentatively identified only in −ve mode. MS/MS analysis proved that the −ve mode was more sensitive to salvianolic acid isomers (Table 4). The main characteristic products of salvianolic acid isomers were [M-H-198]^−^, [M-H-180]^−^, indicating the loss of danshensu and caffeic acid [17].

Isosalvianolic acid B and salvianolic acid L have the same *m*/*z* but differ in their elution time (Appendix A); they were tentatively identified by comparing the retention time and fragmentation pattern with the data published by [24]. The source of salvianolic acid L was an oxidative cyclization process that produced a 1, 2-dihydronaphthalene ring structure, which resulted in the condensation product of two rosmarinic acid molecules [25].

Phenolic acid glycosides (**22**–**30**) were tentatively identified due to glycosidic link cleavage, which resulted in the *m*/*z* of the phenolic acid, followed by neutral losses of H_2_O and CO_2_, as in the current instance of mono- and di-*O*-glycosides phenolic acid.

Total ion current chromatograms in both −ve and +ve ionization modes (Figure 5 and Figure 6) showed the first detected *m*/*z* from ethanolic extract of *Salvia hispanica* seeds for vanillic acid *O*-hexoside (**23**), ferulic acid *O*-hexoside (**24**), protocatechueic acid-*O*-hexoside (**25**), caffeic acid *O*-hexoside, hydroxybenzoic acid-*O*-hexoside (**27**), and caffeic acid di-*O*-rhamnoside (**30**) were 355.0736, 315.0720, 341.0981, 299.0774, and 471.1412, respectively; their characteristic fragment ions were at *m*/*z* 193.0508 [M-H-162(hex)]^−^, 153.0196 [M-H-162(hex)]^−^, 179.0345 [M-H-162(hex)]^−^, 135.0447 [M-H-162(hex)-CO_2_]^−^, 164.0762 [M-H-caffiec]^−^, 137.0236 [M-H-162(hex)]^−^, 145.9292 [M-H-146(rham)-179(Caffoyl)]^−^, and 324.9018 [M-H-146(rham)]^−^, respectively.

In Egyptian *Salvia hispanica* species, a single mono-*C*-glycosidic linkage of phenolic acids was tentatively identified for the first time, with the molecular formulas C_13_H_16_O_9_ and (Rt, 1.044 min) as protocatechueic acid-*C*-hexoside (**22**). It was confirmed based on its MS fragment data [M-H-]^−^ at 271.0386 [M-H-CO_2_]^−^, 195.0514 [M-H-120]^−^, and 225.0031 [M-H-90]^−^. Caffeic acid, ferulic acid, rosmarinic acid, coumaric acid, vanillic acid *O*-hexoside, ferulic acid *O*-hexoside, protocatechueic acid *O*-hexoside, and caffeicacid *O*-hexoside were the most abundant phenolic acids that appeared in Egyptian *Salvia hispanica* species and their structures (Figure 7) were tentatively identified by HPLC/HR-QTOF-MS/MS.

#### 2.4.2. Flavonoids Glycosides

MS cleavage behaviors of flavonoid glycosids have been published previously, and they exhibited characteristic fragments resulting from serial losses of sugar ions as (CH_2_O), in addition to flavones’ overall fragmentation patterns like repeated neutral losses and retro-Diels-Alder (RDA) cleavages in *C*-glycosidic linkage. Flavonoid-*O*-glycocsides include RDA cleavage of ring C, multiple neutral losses, and the successive removal of sugar residues [26]. The fragmentation pattern for MS^2^ clearly distinguishes between these two flavonoid classes. Since the carbon–carbon link of *C*-glycosylated flavonoids is resistant to disruption, the principal cleavages take place at the sugar bonds. However, the sugar moieties of *O*-glycosylated flavonoids are rapidly lost due to neutral losses [27]. Flavonoids were evaluated by systematically matching them to our internal library and by contrasting them with MS data found in the literature.

#### 2.4.3. Flavonol Derivatives

Five isomers (Figure 5) were classified as flavonol-di-*O*-glycosides with the same *m*/*z* at (593.1432/593.1189/593.3003/593.0215/593.2627), different retention times at (6.475/6.530/8.628/14.577/14.629 min), and other chemical formulas: C_28_H_32_O_15_, C_26_H_26_O_16_, and C_27_H_30_O_15_. They generated the same deprotonated aglycone fragment at *m*/*z* 284.0359/284.0324/285.0410/285.0348/284.5016 and neutral loss of di-sugars, proving peaks (**35**–**38**), 41, and 52 were produced from kaempferol and isorhamnetin aglycons. The MS/MS demonstrated their product ions at *m*/*z* 447.1335 [M-H-146(rham]^−^, 431.0731 [M-H-162(gl)]^−^, 285.0410, and 284.0324 for kaempferol-3-*O*-neohesperidoside (**34**); 447.0907 [M-H-146(rham]^−^, 431.1450 [M-H-162(gl)]^−^, and 284.0359 [M-H-162(gl)-146(rham]^−^, for kaempferol-7-*O*-neohesperidoside (**35**); 461.2058 [M-H-132(pent)], and 285.0268 [M-H-132(pent)-176(gluc)]^−^ for Kaempferol 3-*O*-pentoside-7-*O*-glucuronide (**38**); 416.8290 [M-H-176(gluc)]^−^, and 285.0348 [M-H-176(gluc)-132(pent)]^−^ for kaempferol 3-*O*-glucuronide-7-*O*-pentoside (**41**); 460.9255 [M-H-132(pent)]^−^, 447.1633 [M-H-146(rham)]^−^, and 299.5170 [M-H-132(pent)-146(rham)-CH_3_]^−^ for isorhamnetin 3-*O*-pentoside7-*O*-rhamnoside (**52**).

The existence of two high-abundance primary ions in the MS^2^ spectrum proved the position of each sugar linkage to aglycone. The high intensity of the predominant product ion at 284.0324 proved the 3-OH glycosylation site, as in kaempferol-3-*O*-neohesperidoside (**34**). The presence of a central daughter ion at *m*/*z* 285.3020 and a weak ion at *m*/*z* 284.0359 demonstrated 7-OH-glycosylation substitution, as in kaempferol-7-*O*-neohesperidoside (**35**); in addition to kaempferol-3-*O*-α-L-rhamnoside (**33**) at *m*/*z* 431.1766, and (Rt, 1.978 min). In the same way, others were tentatively recognized for kaempferol, quercetin, myricitin, and their derivatives [28,29], their structures are drawn as in (Figure 7).

Two peaks tentatively identified in −ve mode more than +ve mode possessed the same *m*/*z* (447.2838/447.2007), different molecular formulae (C_21_H_20_O_11_/C_22_H_22_O_12_), and different elution times (Rt, 6.698/7.353); they were tentatively proposed to be kaempferol-3-*O*-hexoside (**36**), and the highly concentrated component isorhamentin-3-*O*-hexoside (**65**) due to the loss of hexose and methyl moieties at *m*/*z* 285.0440 [M-H-162(gl)] and 301.3301 [M-H-162(gl)-CH_3_] [30].

#### 2.4.4. Flavone Derivatives

Luteolin4′-*O*-hexoside (**64**) and luteolin-7-*O*-hexoside (**67**) had the same *m*/*z* (447.2077/447.0972) and were detected based on their daughter ions, produced due to loss of the hexose moiety at *m*/*z* 284.0303/284.0327, and 285.0370/285.0461 [M-H-162(gl)]^−^. The site of 3-OH linkage determined by the intensity of 285.0370 [M-H-162(gl)]^−^ is lower than that of the abundance product ion at *m*/*z* 284.0303 [M-H-162(gl)]^−∙^. The presence of the dominating ion at *m*/*z* 285.0461 [M-H-162(gl)]^−^ in compound 67 is higher than the weak ion at *m*/*z* 284.0327 [M-H-162(gl)]^−∙^, suggesting the glycoside is in the 7-OH position.

Two isomers were classified as flavone-diglycosides with the same *m*/*z* at (593.2098/593.2286) and different retention times at (Rt, 5.328/6.052). They were tentatively established based on their intensive fragments at *m*/*z* 473.1159 [M-H-120]^−^, 431.1121 [M-H-120-132(pent]^−^, 298.0425 for diosmetin 8-*C*-hexoside-7-*O*-pentoside (**59**) and 473.2267 [M-H-120]^−^, 269.1150 [M-H-324(digl)]^−^ for vitexin-7-*O*-hexoside(**62**). The loss of 120 and 90 amu showed a di-hexose-*C*-glycosidic connection. The high intensity of a main product ion at [M-H-120]^−^ confirmed hexose at the 8-*C*-position of aglycone in the A-ring. The loss of [M-H-90]^−^ with high intensity confirmed that hexose is at the 6-*C*-position [31]. The characteristic daughter ions at *m*/*z* 341.0640 [M-H-120]^−^, 370.8851 [M-H-90]^−^, 299.0790, 286.8904, 301.1072, 343.1250, and 283.1101 that matched with the high abundance ion at (Rt, 7.495 min) confirmed peak 69 as Diosmetin-6-C-hexoside.

#### 2.4.5. Flavanones

One flavanone compound was tentatively characterized for the first time at (Rt, 8.263 min), and *m*/*z* 593.1828 was suggested as isosakuranetin-7-*O*-neohesperidoside (**75**). Daidzein-8-*C*-hexoside (Puerarin) is an isoflavone identified tentatively in both +ve and −ve modes. It was reported for the first time in an ethanolic extract of *Salvia hispanica* seeds based on its molecular formulae (C_21_H_20_O_9_) and characteristic fragments at *m*/*z* 352.8659, 327.1397, 303.1324, 329.1250, 253.1495, 237.0435, and 255.1210.

#### 2.4.6. Flavans-3ol

Peaks 77 and 78, which display (+) catechin and (+) gallocatechin, are classified as flavan-3-ol structures with the molecular formulas C_15_H_14_O_6_ and C_15_H_14_O_7_. They produced a comparable fragmentation pattern in the MS/MS spectra at *m*/*z* 289.1634 and 305.0738 by RDA and product ions at *m*/*z* 271.1141 [M-H-H_2_O]^−^, 245.0942 [M-H-CO_2_]^−^, and 179.0606 [M-H-125(C_6_H_5_O_3_^•^)]^−^, respectively [32].

#### 2.4.7. Ellagic Tannins

In the MS/MS spectrum, peaks 79 and 80 at *m*/*z* (433.2188, 463.6302) and generated product ions at *m*/*z* 301.0137 and 299.9887 correspond to the loss of pentose and hexose units, proving the metabolites as ellagic pentoxide and ellagic acid-hexoside.

#### 2.4.8. Anthocyanins

Two isomers of anthocyanins were tentatively assigned as delphinidin-3-*O*-glucopyranosid (**81**) and delphinidin-3-*O*-(6″-*O*-*α*-rhamnopyranosyl-*β*-glucopyranoside) (**82**). They displayed generated ions in [M-H] at *m*/*z* 301.0343 [M-H-162(gl)]^−^ for peak (**81**), 563.3040 [M-H-146(rham)]^−^, and 447.1565 [M-H-162(gl)]^−^ for peak **82**, respectively [33,34].

#### 2.4.9. Coumarin

Two peaks of coumarin compounds were tentatively assigned as esculin (**97**) and daphnetin-7, 8-dihydroxycoumarins (**98**). They displayed [M-H]-ions at *m*/*z* 339.0652 and 177.0187 [35].

#### 2.4.10. Lignans and Stilbenes

Lignans were reported as bioactive compounds with strong anticarcinogenic, anti-inflammatory, and antioxidant qualities, as characterized by LC-MS/MS. Compound 86 in [M+H]^+^ mode at *m*/*z* 359.2283 was identified as pinoresinol that showed daughter ions at *m*/*z* 325.12159 [M+H-CH_3-_H_2_O]^+^, 329.00109 [M+H-2CH_3_]^+^, 313.2013, 155.0931, 297.11529 [M+H-2OCH_3_]^+^ with the neutral loss of CH_3_(15 Da),OCH_3_(31 Da), and H_2_O (18 Da) [36]. One stilbene metabolite (**85**) with the chemical formula C_20_H_22_O_9_, time elution (Rt, 8.163 min), and generated ions at *m*/*z* 336.8967 [M-H-4OH]^−^, 296.9058 [M-H-C_6_H_5_O_2_•]^−^, 272.0836 [M-H-C_8_H_7_O_2_•]^−^, 243.1549 [M-H-162(gl)]^−^ was tentatively established in −ve mode as E-3,4,5′-trihydroxy-3′-glucopyranosylstilbene.

#### 2.4.11. Alkaloids

A total of three alkaloid components were determined using the MS/MS fragment ions by comparing the spectra of reference substances in +ve mode with published data [37]. Two ions in high abundance at different molecular formulae (C_20_H_26_NO_4_/C_20_H_20_NO_4_), different elution times (Rt, 15.398/15.459), and different *m*/*z* (339.1709/357.1111) were characterized as jatrorrhizine, and menisperine. The appearance of daughter ions at *m*/*z* 309.1431 [M-H-OCH_3_]^−^, 293.2282 [M-H-OCH_3_-CH_3_]^−^, 291.2018, 275.2087 [M-H-3CH_3_-H_2_O]^−^, 274.9289 [M-H-2OCH_3_]^−^, 279.2315, 321.2729, and 276.9100. [M-H-2OCH_3_]^−^ corresponded to the peak (**100**) of jatrorrhizine. The generated ion at *m*/*z* 311.0995 [M-H-OCH_3_-CH_3_]^−^ was characteristic of menisperine (**101**).

#### 2.4.12. Diterpenoids

Highly concentrated diterpenoid metabolites were characterized by comparing them with MS data from the literature and systematically matching them to our library. Trijuganone C and tanshinone I were the most highly concentrated diterpenes in chia seeds hydroethanolic extract and are represented in red color (Figure 8).

Sixteen diterpenoid compounds were tentatively identified only in positive mode (+ve) (Table 4) as follows. Peak (**105**) with (Rt, 15.169 min) showed a protonated precursor ion at *m*/*z* 277.1856 [M+H]^+^, and daughter ions at *m*/*z* 259.9341 [M+H-CH_3_]^+^, 246.9221 [M+H-2CH_3_]^+^, 137.0590 [M-H-C_11_H_8_^2^•]^−^, and 161.1012 [M+H-CH_3_-C_9_H_8_^2^•]^+^ were specific and characteristic to tanshinone I. A precursor ion at *m*/*z* 295.0962 [M+H]^+^ and fragment ions at *m*/*z* 277.1797 [M+H-H_2_O]^+^, 249.1854 [M+ H-18-28]^+^, 231.1707, [M+H-H_2_O-CO-H_2_O]^+^, 161.0965, and 137.0587 were both present in Compound **106** with (Rt, 15.206 min). As a result, the compound was determined to be tanshinone IIA. Compound **115** with (Rt, 23.697 min) was detected to be 15, 16-dihydrotanshinone based on the presence of a precursor ion at *m*/*z* 279.2340 [M+H]^+^ and a fragment ion at *m*/*z* 264.0875 [M+H-CH_3_]^+^, 170.9375, and 233.5410 [38].

Three precursor ions having the same molecular weights at *m*/*z* (313.1525/313.1334/313.2731) [M+H]^+^ were detected at different time elution (Rt, 15.238/14.743/24.132 min)in peaks **107**, **108**, and **117**, respectively. The specific daughter ions for tanshindiol C (**107**) were due to a loss of 18 amu followed by 28 amu at *m*/*z* 295.0780 and 267.2577, while fragments at *m*/*z* 295.0947 [M+H-H_2_O]^+^, 281.1056 [M+H-CH_2_OH]+, and 277.2162 were recognized for hydroxycryptotanshinone (**108**). The ions at *m*/*z* 297.9810 [M+H-CH_3_]^+^, 254.4611 [M+H-C_3_H_7_O•]+, 149.0223, and 95.0850 [M+H-C_12_H_11_O_4_•]^+^ are typical of hydroxy tanshinone VI (**117**) [39].

In the positive LC-MS/MS mode, a precursor ion at (Rt, 23.508 min) was detected at *m*/*z* 357.2975 [M+H]^+^ in compound 116, and the fragment ions appeared at *m*/*z* 339.2889 [M+H-H_2_O]^+^, 290.5238 [M+H-C_4_H_3_OH]^+^, and 181.1620. As a result, compound 116 was recognized as salviacoccin. The sesquiterpene compound was identified as *β*-Caryophyllene (**112**) based on its molecular ion peak at *m*/*z* 204.1768 and characteristic fragments at *m*/*z* 175.6093, 149.0242, 109.952, and 93.0347.

### 2.5. Investigation of the Chemical Profile of the Petroleum Ether Extract of Salvia hispanica Seeds by GC–MS

The fatty acid (FAME) composition of chia seeds analyzed by GC–MS is presented in Table 5.

### 2.6. Investigation of the Chemical Profile of the Petroleum Ether Extract of Salvia hispanica Seeds by HPLC/QTOF-MS/MS in Both Negative (−ve) and Positive (+ve) Ionization Modes

Fatty acids perform a variety of biological tasks, including cytotoxic action. Hence, our investigation focused on identifying them from the petroleum ether extract [40].

Numerous fatty acids were found as noticeable peaks in the qualitative studies of the petroleum ether extract of *Salvia hispanica* seeds performed by HPLC/QTOF-MS/MS, as shown in (Figure 7); the most sensitive mode for fatty acids is ionization in negative mode (−ve). Fifteen least polar metabolites (fatty acids) were tentatively assigned from their high-resolution masses and predicted formulae in peaks **2**,**3 and** (**6**–**18**) (Table 6 and Figure 9).

The MS spectra of more abundant saturated and unsaturated fatty acids with negative ion mode (−ve), i.e., stearic acid (**5**), eicosaenoic acid (**17**), palmitic acid (**16**), and hydroxy-oxohexadecanoic acid (**15**), were straight forward to tentatively recognize with a characteristic precursor at *m*/*z* of 283.2637, 309.1000, 255.2306, and285.2703 and a predicted molecular formula of C_18_H_35_O_2_, C_20_H_37_O_2_, C_16_H_31_O_2,_ and C_16_H_29_O_4_, respectively.

Additionally, the investigation of the petroleum ether extract suggested a variety of other hydroxylated unsaturated fatty acids, namely dihydroxy-octadecenoic acid (**8**), hydroxy-octadecadienoic acid (**10**), hydroxy-octadecatrienoic (**11**) acid, dihydroxy-octadecadienoic acid (**12**), hydroxy-octadecenoic acid (**14**), and hydroxy-oxohexadecanoic acid (**15**). These give a mass at *m*/*z* 313.2400, 295.2616, 293.1663, 311.1680, 297.2771, 285.2703, with characteristic product ions resulting from the loss of mono or di molecules of water at *m*/*z* 295.2256 [M-H-H_2_O]^−^, 277.043 1 [M-H-H_2_O]^−^, 275.1851 [M-H-H_2_O]^−^, 293.221 [M-H-H_2_O]^−^ and 278.9026 [M-H-H_2_O]^−^; with a difference of 2 amu in a mass between the peaks for proving the presence of an extra double bond.

## 3. Discussion

The current study was performed to evaluate the ability of chia seed extracts; hydroethanolic, and petroleum ether extract to treat lung cancer. 4-(Methylnitrosamino)-1-(3-pyridyl)-1-butanone (NNK)-mediated lung carcinomas in Wister albino rats were used as an experimental model. NNK administration stimulated inducible lung cancer biomarkers, including (i) ICAM-1 level, c-MYC, and MMP9 genes, (ii) inflammation, (iii) angiogenesis, and (iv) proliferation. Meanwhile, NNK inhibited preventive biomarkers of lung cancer, including antioxidant enzymes and apoptosis. On the contrary, the oral administration of chia extracts for five months recorded promising results. Chia materials caused considerable improvement in all parameters of NNK-mediated lung cancer in rats.

The obtained results agreed with Güzel et al. [41], who reported that chia seed ethanol extract showed an antiproliferative effect against A549 human lung cancer cell lines using the MTT method. El Makawy et al. [42] explained the anti-breast cancer effect of quinoa and chia oil nanocapsules that was via the control of PIK3CA and MYC expression, anti-inflammation, and cell proliferation inhibition. Ortega and Campos [43] reported the anticancer effect of chia seed oil in breast cancer cell lines (MCF-7) using the MTT method. Shaer and Al-Abbas [44] demonstrated the anticancer effect of the chia seeds crude extract nanoparticles on MCF-7 breast cancer cells. El Makawy et al. [45] reported the suppressive role of nano-encapsulated chia oil against DMBA-induced breast cancer through oxidative stress repression and tumor gene expression modulation in rats.

The following section provides an explanation of the efficacy of the chia seed extract approach in the treatment of lung carcinoma induced by NNK in rats. Chia seeds struggle with NNK-induced lung carcinoma in rats through (i) inhibition of c-MYC and MMP9 gene expression, (ii) decreased proliferation, (iii) increase in apoptosis, (iv) reduction in angiogenesis, (v) suppression of IL-6 and IL-1β levels, and (vi) increase in non-enzymatic (GSH content) and activation enzymatic (GR, GST, and GPx) activities.

Chia materials inhibited c-MYC and MMP9 gene expression. It has been proposed that the up or down-regulation of MYC mRNA levels controls the entering and exiting of the cell cycle. MYC stimulates the cell cycle mainly through the repression of cell cycle inhibitors. MYC stimulates the cell cycle progression through the regulation of many genes related to cell cycle control [45,46]. MYC regulation and transcriptional activity are essential for normal biological processes such as cell division and apoptosis. The three members of the MYC oncogene family, c-MYC, MYCN, and MYCL, encode c-MYC, N-MYC, and L-MYC, respectively. MYC oncoproteins belong to the super-transcription factor family, which might be responsible for at least 15% of all transcription occurring in the genome. Numerous biological processes, including immunological monitoring, cell survival, differentiation, and proliferation, are regulated by the principal downstream effectors of MYC [47]. The aberrant expression of proto-oncogenes, particularly the overexpressed c-MYC gene in 70% of cancer cells, is intimately linked to the development of tumors. In human solid tumors, c-MYC is often activated and functions as a critical node of the downstream signaling network. It regulates biological processes related to tumors, including cell proliferation and apoptosis [48].

Additional evidence is provided by the rise in double-strand breaks (DSBs) that accompany c-Myc overexpression, which is experimentally detected by an increase in H2AX foci, a histone H2A protein family variation, and the DSB biomarker [49]. It is also believed that c-MYC can cause double-strand breaks (DSBs) by raising reactive oxygen species (ROS) and interfering with non-homologous end joining, which is the primary mechanism responsible for DSB repair in mammals. Additionally, c-MYC suppresses the expression of proteins known as bridging integrator 1 (BIN1), which monitor DNA replication errors and ordinarily cause apoptosis or senescence. In addition, c-MYC regulates anti-apoptotic proteins such as BCL2 that can prevent the propagation of cells with double-strand brakes (DSBs) [50].

Other important mechanisms, such as angiogenesis, immune system evasion and regulation, and remodeling of extracellular matrix (ECM) components, are also required for effective metastasis. Adhesion to ECM components (such as collagen, elastin, and fibronectin) must be overcome for tumor cell motility and dissemination. c-MYC up-regulates a variety of proteases that can cleave these ECM molecules, namely matrix metalloproteinases (MMPs) and cathepsins [51]. The main MMP family member that breaks down collagen and fibronectin to promote tumor migration is MMP-9. MMP-9 secretion has been demonstrated to be driven by c-Myc expression in tumors and tumor-associated macrophages (TAMs), despite the fact that it is not a direct transcriptional target. Recently, Hu and Lu [52] demonstrated that c-MYC activates the long noncoding RNA brain cytoplasmic RNA 1 (BCYRN1), which in turn promotes metastasis in non-small-cell lung cancer (NSCLC) by secreting MMP9 and MMP13. Pello et al. [53] used a genetically engineered mouse model to delete c-MYC in myeloid precursor cells and demonstrated that this inhibits the maturation of TAMs, decreases MMP9 expression, and limits tumor metastasis. In the present study, both hydroethanolic and ether chia seed extracts significantly reduced c-MYC and MMP9 expression by about 96 and 69% for hydroethanolic extract and by about 80 and 60%for ether extract, respectively, compared to the NNK control.

Chia seed extracts reduced angiogenesis: The creation of new blood vessels is known as angiogenesis. The progression of cancer is mainly dependent on angiogenesis since solid tumors require blood flow to expand beyond a few millimeters in size. The ensuing new blood vessels “feed” the expanding tumors with nutrients and oxygen, causing the tumor to grow larger and the cancer cells inside to infiltrate surrounding tissue, spread throughout the body, and create new cancer cell colonies known as metastases. One powerful angiogenic agent is vascular endothelial growth factor (VEGF) [54]. Lung cancer is known to overexpress VEGF, which is linked to increased angiogenesis, tumor growth, and metastasis. Numerous trials have shown that anti-VEGF therapy enhances both overall and progression-free survival in patients with advanced non-small-cell lung cancer. In the realm of angiogenesis in lung cancer, new anti-angiogenic medications and combination therapies that target multiple pathways involved in tumor angiogenesis have recently been studied [55]. In the present study, both chia seed ether and alcohol extracts significantly reduced VEGF content by about 19.99 and 22.22% compared to the lung cancer control.

Chia seed extracts decreased proliferation: Cancer is essentially an illness caused by abnormally high cell proliferation. In fact, the majority of cancer treatments used today aim to lower the tumor cell count and stop it from growing further [56]. The proliferation biomarker Ki67 is a protein that is exclusively present in proliferating cells. Ki67 protein is not seen in quiescent (resting) cells (G0), but it is present in all active phases of the cell cycle (G1, S, G2, and mitosis). As a cell advances through the S phase of the cell cycle, the amount of Ki67 protein within the cell increases noticeably. A high Ki67 proliferation index indicates a high rate of cell division and an increased risk of cancer growth and metastasis [56]. Tabata et al. [57] reported that adjuvant chemotherapy is more beneficial for patients with ER-positive breast cancer who belong to a high proliferative subgroup, as shown by Ki67 [58]. In the present study, both chia seed ether and alcohol extracts significantly reduced Ki67 content by about 29.35 and 35.27%, compared to the lung cancer control.

Chia seed extracts increased apoptosis: Most often, a variety of pharmacological treatments cause cancer cells to undergo apoptosis when they die [59]. Cancer’s intrinsic and extrinsic apoptotic pathways are frequently inhibited by a variety of strategies, such as the over and under-expression of pro- and anti-apoptotic proteins [60]. The permeability of the outer mitochondrial membrane and the on/off intrinsic apoptotic pathway are both regulated by BCL2 family proteins. According to specific theories, the BCL2 protein inhibits the depolarization of the mitochondrial membrane potential (MMP), which reduces the activation of downstream apoptotic molecules such as cytochrome c, AIF, and Smac/Diablo [61]. Moreover, overexpressing BCL2 can decrease the synthesis of lipid peroxide and the production of oxygen free radicals. Given that BCL2 may reduce the transmembrane flow of calcium ions, it is plausible that it regulates apoptosis via calcium channels. Apoptotic factors are accumulated in the endoplasmic reticulum, where they release Ca2+ and activate the precursor caspase-12. Activation of caspases-9 and -3 subsequently results in apoptosis [62]. In the present study, both hydroethanolic and ether extracts of chia seeds significantly reduced Bcl2 content by about 52 and 56% compared to the lung cancer control.

Chia seed extracts suppress inflammation. Chronic inflammation develops in the lungs when acute inflammation is not controlled. When smoking or breathing in harmful substances from cigarette smoke, macrophages frequently release TNF-α, IL-6, IL-1β, and IL-8 [63]. According to Sionov et al. [64], neutrophils in chronic inflammation either boost tumor-related inflammation, angiogenesis, and metastasis or limit tumor growth by expressing cytotoxic and antitumor mediators. Airway inflammation was greatly reduced, and lung tumor cell proliferation and angiogenesis were inhibited by inhibiting IL-6 using a monoclonal antibody [65]. Following the activation of an inflammatory cytokine known to be pro-tumor, this infamous inflammatory cytokine also stimulates the formation of blood vessels and lymphatic angiogenesis. Tumor cells may migrate across great distances only with the help of new blood vessels, which is made possible by angiogenesis. TNF-α and IL-1β are examples of pro-inflammatory cytokines that activate the NF-κB pathway, a crucial regulator of cell growth and proliferation. On the other hand, ongoing NF-κB activity in TME stimulates tumor cell invasion and EMT while also inducing angiogenesis and death [66]. In the current investigation, compared to a lung cancer control, chia seed oil and hydroethanolic extract significantly decreased the expression of IL-6 and IL-1β (by around 10.8 and 13.17% for ether extract and 4.25 and 11.20% for alcohol extract, respectively).

Chia materials activate the antioxidant system either non-enzymatically (GSH content) or enzymatically (GR, GST, and GPx, activities): Oxygen and nitrogen free radicals can also worsen cellular damage through a variety of mechanisms, including (i) DNA damage leading to breaks and genetic alterations; (ii) lipid degeneration accompanied by pro-inflammatory and vasoactive molecules like thromboxane formation; (iii) protein oxidation (primarily in sulfhydryl groups) influencing protein expression and triggering protease release, antioxidant, and non-protease activation; lipid peroxidation, hence, has the potential to affect anti-protease and surfactant surfaces on cells, immunological modulation, extracellular matrix, alveolar repair, and respiratory protective factors. The defensive mechanisms against oxidative stress caused by free radicals consist of four main processes: (i) repair mechanisms, (ii) preventative mechanisms, (iii) physical defenses, and (iv) enzymatic and non-enzymatic antioxidant defenses. Enzymatic antioxidant defenses include glutathione peroxidase (GPX), SOD, and catalase. Non-enzymatic antioxidants include flavonoids, carotenoids, vitamin E (a-tocopherol, GSH), and vitamin C (ascorbic acid [67]. The chia extracts in the present study significantly reduced oxidative stress biomarkers (MDA) and activated antioxidants, either non-enzymatic (GSH) or enzymatic (GR, GST, and GPx).

Finally, the authors suggested that chia seed alcoholic or ether extracts fought NNK-induced lung cancer in rats through (i) reduction in c-MYC and MMP9 gene expression, (ii) reduction in angiogenesis, (iii) increase in apoptosis concurrent with decreased proliferation, (iv) suppression of inflammation, and (v) activation of the antioxidant system, either non-enzymatic or enzymatic.

The chia seed ethanolic extracts exhibited these characteristics due to their chemical composition, which includes considerable quantities of secondary metabolites identified by LC-Mass. Many anticancer components were found in chia extracts. The significant components of chia alcohol extract are caffeic acid, vanillic acid hexoside, kaempferol-3-*O*-glucuronide, taxifolin, and two diterpenes: Trijugunone C (Figure 8) and tanshinon, as well as one alkaloid, Jatrorrhizine. The anticancer effects of all these components are demonstrated below.

Several studies demonstrated the anticancer capacities of caffeic acid. According to a different study, 600 mM of caffeine significantly increases the rate of apoptosis in H1299 and A549 cells. It may also have an effect on regulating and enhancing PTX-mediated NSCLC cell death in both in vitro and in vivo MAPK signaling [68]. The caffeic acid exposure also increased the expression of the anti-apoptotic proteins survivin and Bcl-2 in another non-small-cell lung cancer cell line, A549 [69]. Nevertheless, in mouse lung adenocarcinoma LA-795 cells, caffeic acid (60 µM) decreased the cell viability to approximately 50% [70]. Caffeic acid phenyl ester inhibits the growth of tumor cells using oxidative stress pathways connected to p53-independent pathways and decreases the generation of intracellular hydrogen peroxide (H_2_O_2_) in A549 cells [71]. Ulasli et al. [72] revealed that caffeic acid phenyl ester up-regulates the key upstream signaling factors, ultimately increasing their regulatory p53 levels and affecting the induction of G2/M cell cycle arrest and apoptosis.

Numerous studies have shown that vanillic acid and its glycosides have anticancer properties. Vanillic acid interfered with tube formation and reduced the production of the VEGF. Vanillic acid strongly suppressed the growth of human colon cancer HCT116 cells and triggered G1 phase arrest. Experiments in a xenografted tumor model showed that vanillic acid therapy significantly inhibited tumor growth [73]. Vanillic acid (75 mg/kg body weight) demonstrated an antiproliferative effect in rats exposed to diethyl nitrosamine and 1,2-dimethylhydrazine-induced liver and colon carcinogenesis, as seen by the decrease in Cyclin D1 expression. The overexpression of Caspase-3 and Bad levels and the down-regulation of Bcl-2 levels may be the cause of the apoptotic activity [74].

Kaempferol and its primary plasma conjugate, kaempferol-3-*O*-glucuronide, showed inhibitory potency on protein kinases of HepG2 lysate [75]. Kaempferol-3-*O*-glucuronide isolated from *Calligoum polygonoides* recorded cytotoxic effects against liver cancer (HepG2) and breast cancer (MCF-7) cell lines [76]. Taxifolin (dihydroquercetin) is a flavononol that has antiproliferative properties in two lung cancer cell lines, A549 and H1975, as well as in A549 xenografts. Taxifolin increased the expression of E-cadherin and decreased invasive cells, N-cadherin, and vimentin, suggesting that the epithelial-mesenchymal transition (EMT) was prevented [77]. Taxifolin induces cytotoxicity in colorectal cancer cell lines (HCT116 and HT29) as well as the HCT116 xenograft model. Also, at an inhibitory concentration (IC_50_) value of 0.15 mM and 0.22 mM, respectively, taxifolin treatment led to the suppression of liver cancer growth and migration and triggered apoptosis in HepG2 and Huh7 cell lines. Strong pro-apoptotic and hepatoprotective effects were demonstrated by taxifolin [78].

Tanshinone and Trijuganone C showed significant antiproliferative activities against several human cell lines, including human leukemia cells HL-60, Jurkat, and U937. Also, Trijuganone C recorded cytotoxic effects against colon cancer cell lines DLD-1, COLO 205, Caco-2, and HCT-15. Trijuganone C showed antiproliferation against prostate cancer (PC-3 and LNCap FGC), breast cancer, MCF-7, liver cancer, and HepG2 [79].

Tanshinone I diminished the tumorigenesis and metastasis in CL1-5-bearing severe combined immunodeficient mice and markedly decreased migration, invasion, and gelatinase activity in macrophage-conditioned medium-stimulated CL1-5 cells in vitro [80]. Tanshinone inhibited the proliferation of human endometrial carcinoma HEC-1-A cells due to the induction of apoptosis. Tanshinone I also increased the ROS levels in these cells, which was linked with a reduction in MMP levels [81].

Also, chia seed petroleum ether extract exhibited these characteristics due to its chemical composition, which includes considerable quantities of secondary metabolites identified by GC–MS and LC-Mass, such as β-Sitosterol (43.10%) and linolenic acid (67.48%). β-Sitosterol significantly inhibited the growth of A549 cells without harming normal human lung and PBMC cells. Further, BS treatment triggered apoptosis via ROS-mediated mitochondrial dysregulation as evidenced by caspase-3 and 9 activations, Annexin-V/PI-positive cells, PARP inactivation, loss of MMP, Bcl-2-Bax ratio alteration, and cytochrome c release. Moreover, the generation of ROS species and subsequent DNA strand breaks were found upon β-Sitosterol treatment. Indeed, β-Sitosterol treatment increased p53 expression in A549 cells [82]. In addition, β-Sitosterol effectively suppressed proliferation and promoted apoptosis of lung cancer using the A549 cell line and A549/anlotinib cells [83]. Linolenic acid exerts significant anticancer effects on multiple cancers, including lung cancer. Its various effects include inhibiting proliferation, inducing apoptosis, suppressing tumor metastasis and angiogenesis, and exerting antioxidant effects [84].

Finally, the remarkable anticancer, anti-angiogenesis, antiproliferation, anti-inflammatory, and antioxidant effects of chia seed extracts may be attributed to a synergistic effect between the mentioned compounds.

## 4. Materials and Methods

### 4.1. Chemicals

4-(Methylnitrosamino)-1-(3-pyridyl)-1-butanone (NNK) is a nitrosamine present in smokeless tobacco that produces cancer in rats and hamsters (Molecular Weight: 207.23, Molecular Formula: C_10_H_13_N_3_O_2_, |CAS 64091-91-4|SCBT—Santa Cruz Biotechnology Company, Heidelberg, Germany). All chemicals and solvents used are analytical grades. Reagents for invitro cytotoxicity assay: DMEM (Dulbecco’s modified Eagle’s Medium), fetal bovine serum (FBS), penicillin, and streptomycin sulfate, trypsin-EDTA, and DMSO (Dimethyl sulfoxide). Kits of antioxidant parameters including glutathione (GSH), glutathione reductase (GR), glutathione-*S*-transferase (GST), glutathione peroxidase (GPx), catalase (CAT), and superoxide dismutase (SOD) were purchased from Bio Diagnostic, Egypt. Kits of safety profiles, including liver function and renal functions, were obtained from Bio Diagnostic, Egypt. ELISA kits for interleukin 6 (IL-6) (Cat NO.: SL0547Mo), vascular endothelial factor growth (VEGF) (Cat NO.: SL0314Mo), and IL-1β (Cat NO.: SL0537Mo) were purchased from Sunlong Biotech Co., Ltd., Ping Shui Street, Gong Shu District, Hangzhou, China; email: sales@sunlongbiotech.com. We used QRT-PCR kits for c-MYC, MMP9, and BCL2 genes (RNAeasy mini-Kit (Qiagen, Hilden, Germany) (Cat. NO./ID: 74104), the Revert Aid First Strand cDNA Synthesis Kit (Thermo Scientific, Waltham, MA, USA) (Cat. NO.: K1621), and the Maxima SYBR Green qPCR Master Mix (2X) (Thermo Scientific, USA) (Cat. NO.: K0221).

### 4.2. Authentication of the Plant and Extraction

*Salvia hispanica* (Chia) seeds were obtained from the Agricultural Research Center, Giza, Egypt. Prof. Dr. Abd El-HalimAbd El-Mogali Mohamed, Chief Researcher, Flora &Phytotaxonomy Research Department, HRI, ARC identified the plant. In addition, a voucher specimen (M246) was deposited in the herbarium of the National Research Centre (CAIRC), Giza, Egypt. Seeds (1 kg) were exhaustively extracted with ether (40:60) several times under laboratory conditions by soaking. Then, the same seeds (after drying) were exhaustively extracted with 70% ethanol solution several times under the same conditions. The extracts were concentrated using a Rotary Evaporator and then were lyophilized, and the resulting powder was kept at 20 °C until use.

### 4.3. Studying the Anti-Lung Cancer of Chia Seed Extracts

#### 4.3.1. In Vitro Cytotoxicity Assay

The A549 lung cancer cells were purchased from ATCC (American Type Culture Collection) and preserved in appropriate conditions. A549 cells were cultivated in DMEM (Dulbecco’s modified Eagle’s Medium), which included 10% fetal bovine serum (FBS), 100 U/mL penicillin, and 100 μg/mL streptomycin sulfate at 37 °C in a humidified 5% CO_2_. The cells were digested with 0.025% trypsin-EDTA for passage. Cells in the logarithmic growth phase were used for the experiment. The cytotoxicity of chia seed extracts was assessed using a neutral red uptake test. Several concentrations (25, 50, 100, and 200 µg/mL) of tested extracts were added to continue the culture for 48 h at a cell density of 10^4^ cells/well on 96 well plates, and a neutral red uptake test was done as reported by Repetto et al. [85]. The relationship between the neutral red intensity value and the utilized log concentrations was used to determine the IC_50_ (half maximal inhibitory concentration) of the examined extracts. For the untreated cells, medium was added instead of the tested extracts (negative control). Doxorubicin, Mr. = 543.5, a cytotoxic natural drug, was employed as a positive control, providing 100% inhibition. Chia seed extracts were dissolved in DMSO (Dimethyl sulfoxide), with a final concentration on the cells of no more than 0.2%. Every test was performed at least three times.

#### 4.3.2. In Vivo Study

##### Determination of LD_50_

The LD_50_ of the chia ether extract and 70% ethanolic extract were carried out [86] as per OECD guideline 425 (OECD 2008) for the acute oral toxicity -Up-and-Down- Procedure (UDP) using mice. Doses of extracts ranged from 0.5 g to 6 g/kg with an increased rate of 0.5 g/kg body weight. The extracts were force-fed to mice (5 mice per group) orally using a stomach tube, and control mice were force-fed saline justly. Animals were maintained under observation and checked out for changes and mortality for 48 h. Reminded mice were observed for two weeks. LD_50_, the dose of extracts that killed 50% of animals, was calculated by the number of dead animals in each concentration during the first 48 h using the BioStat program (BioStat 2009 Build 5.8.4.3 # 2021 analyst Soft Inc., Centreville, VA, USA). LD_50_, the chia ether extract, and 70% ethanolic extract were 5 g/kg body weight. The chia ether extract and 70% ethanolic extract were used in this study with a dose of 500 mg/kg, which was calculated as 1/10 of the LD_50_.

##### Lung Cancer Induction

Albino male rats of Sprague Dawley strains (180–200 g, 10 weeks old) were obtained from the animal house of the National Research Centre, Dokki, Cairo, Egypt, and were kept in special plastic cages. The animals were maintained under the laboratory animal conditions as follows (Guide for the Care and Use of Laboratory Animals, 2011) (20–25 °C, 55–65% humidity, and 10–12 h light/dark cycle). Water and food were available ad libitum over five months.

A hundred adult male Sprague-Dawley rats weighing 180–200 g underwent lung cancer induction with 4-(Methylnitrosamino)-1-(3-pyridyl)-1-butanone (NNK). Rats in this group were treated with a dose of 1.5 mg/kg body weight NNK subcutaneously three times weekly for 20 weeks [87]. At the end of 20 weeks, lung cancer was confirmed histologically using five rats.

##### Experimental Layout

Adult male Sprague-Dawley rats were classified into six groups as follows:

Group I: 20 healthy rats received 1mL of distilled water orally for 40 weeks (experimental period) using a gastric tube. This group was kept as the ve- control group.

Group II: 20 healthy rats received 1 mL of distilled water orally for 20 weeks (induction period) and then received chia ether extract (500 mg/kg/day orally for 20 weeks, treatment period). This group was kept as the ve+ of ether extract.

Group III: 20 healthy rats received 1mL of distilled water orally for 20 weeks (induction period) and then received chia ethanolic extract (500 mg/kg/day orally for 20 weeks, treatment period). This group was kept as the ve+ of ethanolic extract.

Group IV: Twenty healthy rats were subcutaneously injected with a dose of 1.5 mg/kg body weight NNK three times weekly for 20 weeks, and then they received 1 mL of distilled water orally for 20 weeks. This group was kept as the cancer control group.

Group V: 20 healthy rats were subcutaneously injected with a dose of 1.5 mg/kg body weight NNK three times weekly for 20 weeks, and then they received chia ether extract (500 mg/kg/day orally for 20 weeks, treatment period). This group was kept as the chia ether extract-treated group.

Group VI: Twenty healthy rats were subcutaneously injected with a dose of 1.5 mg/kg body weight NNK three times weekly for 20 weeks, and then they received chia ethanolic extract (500 mg/kg/day orally for 20 weeks, treatment period). This group was kept as the chia ethanolic extract-treated group.

After 40 weeks, rats were fasted overnight. The fasted rats were anesthetized with ketamine/xylazine (87 and 13 mg/kg, respectively, dissolved in normal saline, and each rat received 0.2 mL/100 g body weight) [88].

Blood samples were collected from the abdominal aorta. Each blood sample was centrifuged at 3500× *g* for 10 min using Sigma Laborzentrifugen (Osterode am Harz, Germany), and the serum was isolated and stored at −20 °C until later analysis. After authorization, vital organs were removed, weighed, and washed with ice-cold saline. The right lung was then immersed in 10% buffered formaldehyde for histopathologic analysis. The left lung was placed at −80 °C for later chemical measurements.

##### Biochemical Assessments

Biochemical analyses were determined in serum spectrophotometrically. Liver functions such as total protein [89], albumin [90], aspartate aminotransferase (AST), and alanine aminotransferase (ALT) [91] were determined. Kidney functions such as urea [92] and uric acid [93] were estimated [94].

Antioxidant biomarkers were estimated in serum and lung homogenates as reduced glutathione (GSH), glutathione reductase (GR), glutathione-S-transferase (GST), and glutathione peroxidase (GPx), and catalase (CAT) were determined according to Griffith [95], Goldberg and Spooner [96], Paglia and Valentine [97], and Habig et al. [98], respectively.

According to the manufacturer’s instructions, using an enzyme-linked immunosorbent assay, serum IL-6, IL-1β, VEGF, MPO, and collagen were determined using ELISA kits.

##### Gene Expression Assessment

Quantitative real-time PCR:

Using the RNAeasymini Kit (Qiagen, Germany) (Cat. NO./ID: 74104), RNA was extracted from rat lung tissues. The concentration and purity of the extracted total RNA were then assessed using the NanoDrop one microvolume UV spectrophotometer (Thermo Fisher Scientific, USA). The Revert Aid First Strand cDNA Synthesis Kit (Thermo Scientific, USA) (Cat. NO.: K1621) was used to convert the RNA from each treatment to first-strand cDNA in accordance with the manufacturer’s instructions. Table 7 lists specific primer sequences. mRNA expression levels of c-MYC, MMP9, and BCL2 genes were normalized with respect to β-actin transcript using Maxima SYBR Green qPCR Master Mix (2X) (Thermo Scientific, USA) (Cat. NO.: K0221) and calculated by 2^−ΔΔCT^ method according to Livak and Schmittgen, [99]. The reaction conditions were as follows: 95 °C for 10 min, 95 °C for 15 s, 55 °C for 30 s, and 72 °C for 30 s, with a total of 40 cycles of amplification. DNA Technology Detecting Thermocycler DT Lite 4S1 (Russia) was used for gene expression quantitation.

##### Histopathological Assessments

Lung specimens were collected and fixed in neutral buffered formalin 10%, routinely processed, and embedded in paraffin wax. Sections of 4–5 µm thickness were prepared and stained with Haematoxylin and Eosin for histopathological examination by light microscope (Olympus CX 41, Tokyo, Japan). Histopathological alterations were graded as (0), indicating no changes, and (1), (2), and (3), indicating mild, moderate, and severe alterations, respectively [100].

### 4.4. Qualitative Analysis of Phytoconstituents in Salvia hispanica (Chia) Seeds Petroleum Ether by GC–MS

#### 4.4.1. Sample Preparation for Lipid Composition GC Analysis

##### Fatty Acid Methyl Esters (FAME)

The lipids of samples were extracted by biphasic3:2Hexane/Isopropanol (*v*/*v*) extraction method; the upper hexane layer contains lipids. Fatty acid methyl esters (FAMEs) are produced by an alkali-catalyzed reaction between fats and methanol in the presence of 2Mpotassium hydroxide and injected in hexane.

##### Unsaponifiable Matter

The sample’s lipids were saponified using ethanolic potassium hydroxide, and the unsaponifiable fraction was extracted using a 1:1 (*v*/*v*) mixture of petroleum ether and ether before being evaporated at 40 °C. Illustration of derivatization: in order to derivatize the sample functional groups to trimethylsilyl groups (TMS) prior to GC analysis, the unsaponifiable fraction extracted with ether and petroleum ether (1:1) was combined with 50 µL of bis(trimethylsilyl) trifuoroacetamide (BSTFA)+ trimethylchlorosilane (TMCS) 99:1 silylation reagent and 50 µL pyridine.

#### 4.4.2. Gas Chromatography

The GC model 7890B from Agilent Technologies was equipped with a flame ionization detector at Central Laboratories Network, National Research Centre, and Cairo, Egypt. A Zebron ZB-FAME column (60 m × 0.25 mm internal diameter × 0.25 μm film thickness) was used to obtain the separation. Hydrogen served as the carrier gas in the analysis with a flow rate of 1.8 mL/min in split-1:50 mode, an injection volume of 1 µL, and the following temperature program: 100 °C for 3 min; rising at 2.5 °C/min to 240 °C and held for 10 min. The injector and detector (flame ionization detector, FID) were held at 250 °C and 285 °C, respectively.

#### 4.4.3. Identification of Chemical Composition of Oil Using Gas Chromatography–Mass Spectrometry Analysis (GC–MS)

The gas chromatograph (7890B) and mass spectrometer detector (5977A) of the Central Laboratories Network, National Research Centre, Cairo, Egypt were both part of the GC–MS system (Agilent Technologies). An HP-5MS column with an internal diameter of 30 m × 0.25 mm and a film thickness of µm was installed in the GC. Hydrogen was employed as the carrier gas for the analysis, with a split of 1:20, a flow rate of 2.0 mL/min, an injection volume of 1 µL, and the following temperature program: 50 °C for five minutes; 5 °C/min rise to 100 °C and held for 0 min; 10 °C/min rise to 320 °C and held for 10 min. The injector and detector were held at 280 °C, 320 °C. Mass spectra were produced by electron ionization (EI) at 70 eV, with a 6min solvent delay and a spectral range of *m*/*z* 25–700. The mass temperature was 230 °C and the Quad was 150 °C. Various ingredients might be identified by comparing the spectrum fragmentation pattern with those found in the Wiley and NIST Mass Spectral Library data.

### 4.5. Qualitative Analysis of Phytoconstituents in Salvia hispanica (Chia) Seeds by HPLC/HR-QTOF-MS/MS

Liquid chromatography-mass spectrometry analysis was used to identify the chemical composition of *Salvia hispanica* (Chia) [18,19]. LC-mass/mass analysis was carried out in the Proteomics and Metabolomics Research Program of the Basic Research Department at the Children’s Cancer Hospital, Cairo, Egypt.

#### 4.5.1. Sample Preparation

A stock solution of the *Salvia hispanica* (Chia) was prepared from 50 mg of the lyophilized *Salvia hispanica* (Chia) dissolved in 1000 µL of the solvent mixture, itself composed of water–methanol–acetonitrile (H_2_O:MeOH:CAN, 50:25:25) (*v*/*v*). Complete solubility of the stock solution was obtained by vortexing the sample for 2 min and ultra-sonicating the sample at 30 kHz for 10 min. An aliquot, 20 µL of the stock solution, was again diluted with 1000 µL of H_2_O:MeOH:CAN, 50:25:25 (*v*/*v*) and centrifuged at 10,000 rpm for 10 min. Finally, 10 µL of stock with a concentration of 2.5 µg/µL was injected. Some 10 µL of reconstitution solvent was injected as a blank sample. The sample was injected in both positive and negative modes.

#### 4.5.2. Instruments and Acquisition Method

The mass spectrometry (MS) was performed on a Triple TOF 5600+ system equipped with a duo-spray source operating in the ESI mode (AB SCIEX, Concord, ON, Canada). The sprayer capillary and declustering potential voltages were −4500 and −80 V in both positive and negative modes. The source temperature was set at 600 °C, the curtain gas was 25 psi, and gas 1 and gas 2 were 40 psi. A collision energy of −35 V (negative mode), a CE spreading of 20 V, and an ion tolerance of 10 ppm were used. The TripleTOF5600+ was operated using an information-dependent acquisition (IDA) protocol. Batches for MS and MS/MS data collection were created using Analyst-TF 1.7.1. The IDA method was used to simultaneously collect full-scan MS and MS/MS information. The technique consisted of high-resolution survey spectra from 50 to 1100 *m*/*z*, and the mass spectrometer was operated in a pattern wherein a 50ms survey scan was detected. Subsequently, after each scan, the top 15 intense ions were selected for the acquisition of the MS/MS fragmentation spectra.

#### 4.5.3. LC-MS Data Processing

MS-DIAL 4.8 open-source software was used for the sample’s non-targeting, small molecule comprehensive analysis. ReSpect positive (2737) and negative (1573 records) databases were used as reference databases according to the acquisition mode. The MS-DIAL output was used to run again on PeakView 2.2 with the Master View 1.1 package (AB SCIEX) for feature (peaks) confirmation based on the criteria, using the total ion chromatogram (TIC). Aligned features had a signal-to-noise ratio greater than 5 and a sample intensity greater than 3.

### 4.6. Statistical Analysis

All results were presented as mean ± SE. The data were calculated using Sigma Plot Ver. 12.5. A statistically significant difference between groups was made using a one-way analysis of variances (one-way ANOVA), followed by the Tukey Kramer multiple comparison test. For data analysis, GraphPad Prism version 8 was used (GraphPad Software Inc., San Diego, CA, USA). A significant difference was set at *p* < 0.05 for all tests.

## 5. Conclusions

In the current study, chia seed ether and alcohol extracts showed anti-lung cancer effects against NNK-induced lung cancer in rats. On the other hand, oral administration of chia extracts for five months significantly reduced lung cancer biomarkers, including the relative weight of the lung, ICAM-1, c-MYC, and MMP9 gene expression, and lung histology was improved considerably compared to the lung cancer control one. Chia seed extracts exhibited inhibitory effects on the levels of VEGF, Ki67, IL-6, and IL-1β biomarkers concurrently with stimulatory effects on the level of BCl2. We can conclude that the potent activities of chia seed against NNK-induced lung cancer may be related to its ability to suppress angiogenesis, inflammation, proliferation, and oxidative stress. Chia seed also induced apoptosis and antioxidant biomarkers. This effective anti-lung cancer activity may be attributed to its content of secondary metabolites identified by LC-Mass. The major components of the alcohol extract, caffeic acid, vanillic acid hexoside, K-3-*O*-glucuronide, taxifolin, and two diterpene compounds such as Trijuganone C and Tanshinone I, demonstrated anticancer components and many anticancer mechanisms in lung cancer both in vitro and in vivo. Additionally, the results obtained from this study, in addition to the fact that chia seeds are an edible food, suggest that chia seeds may be included in a clinical trial, especially since chia showed a high margin of safety in this study.

## Figures and Tables

**Figure 1 pharmaceuticals-17-01129-f001:**
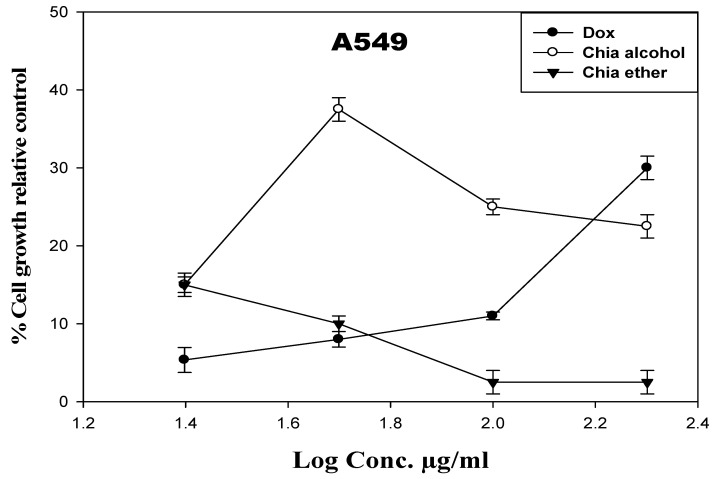
Impact of chia alcohol and chia ether on A549 cells after 48 h.

**Figure 2 pharmaceuticals-17-01129-f002:**
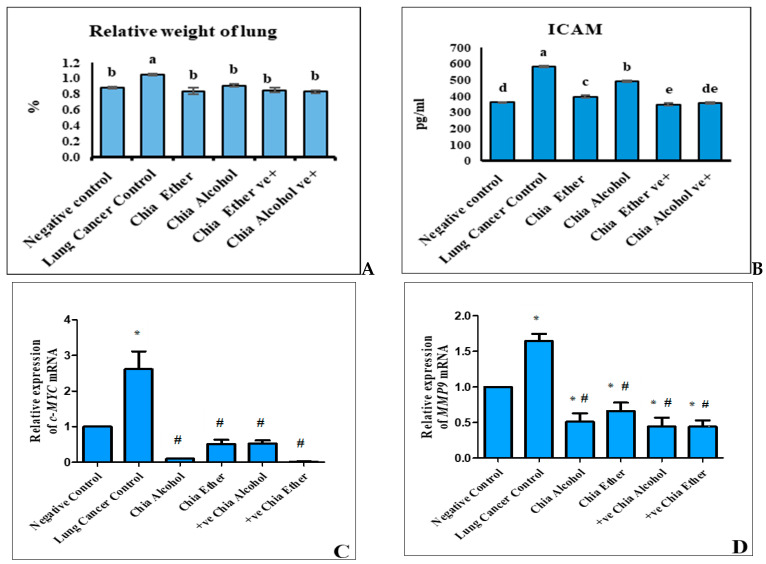
Effect of chia materials on lung cancer biomarkers: (**A**) the relative weight of the lung; (**B**) ICAM level; (**C**) c-MYC expression; and (**D**) MMP9 expression. Each value is the meaning of six replicates. Data were analyzed via the one-way ANOVA test for comparisons among means at *p* ≤ 0.05. In (**A**,**B**), values with the same superscript letter are not significantly different, while values with different letters are significantly different at *p* ≤ 0.05. In (**C**,**D**): * Versus negative control rats at *p* < 0.05. # Versus lung cancer control rats at *p* < 0.05.

**Figure 3 pharmaceuticals-17-01129-f003:**
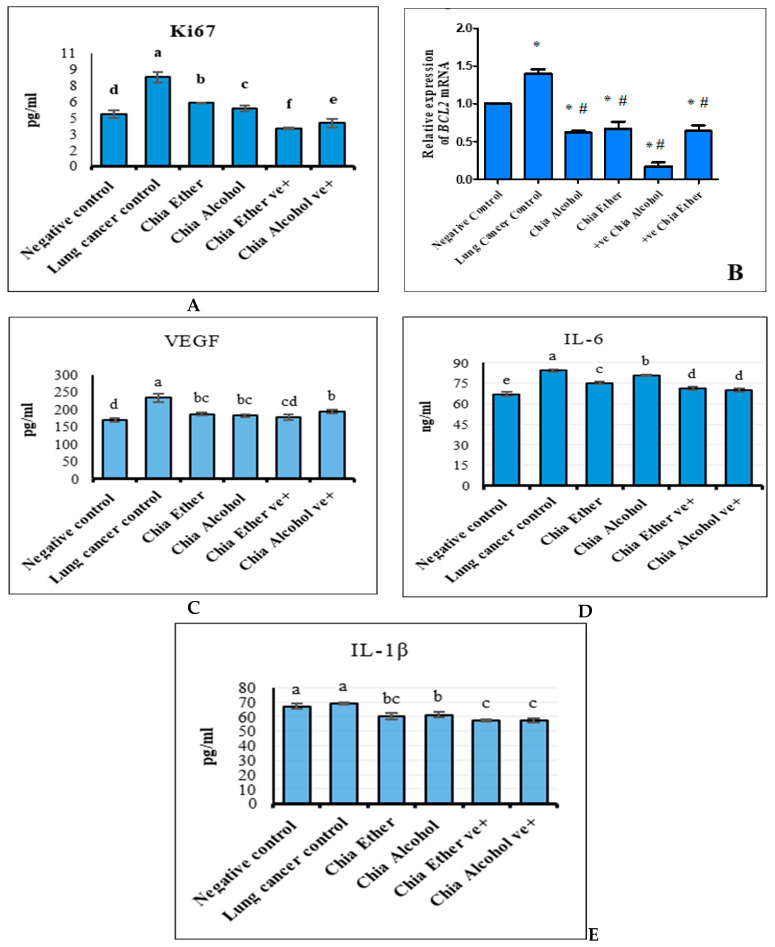
Effect of chia materials on proliferation (**A**), apoptosis (**B**), angiogenesis (**C**), inflammation (**D**,**E**). Each value is the mean of six replicates. Data were analyzed via the one-way ANOVA test for comparisons among means at *p* ≤ 0.05. In (**A**,**C**,**D**,**E**), values with the same superscript letter are not significantly different, while values with different letters are significantly different at *p* ≤ 0.05. In (**B**): * Versus negative control rats at *p* < 0.05. # Versus lung cancer control rats at *p* < 0.05.

**Figure 4 pharmaceuticals-17-01129-f004:**
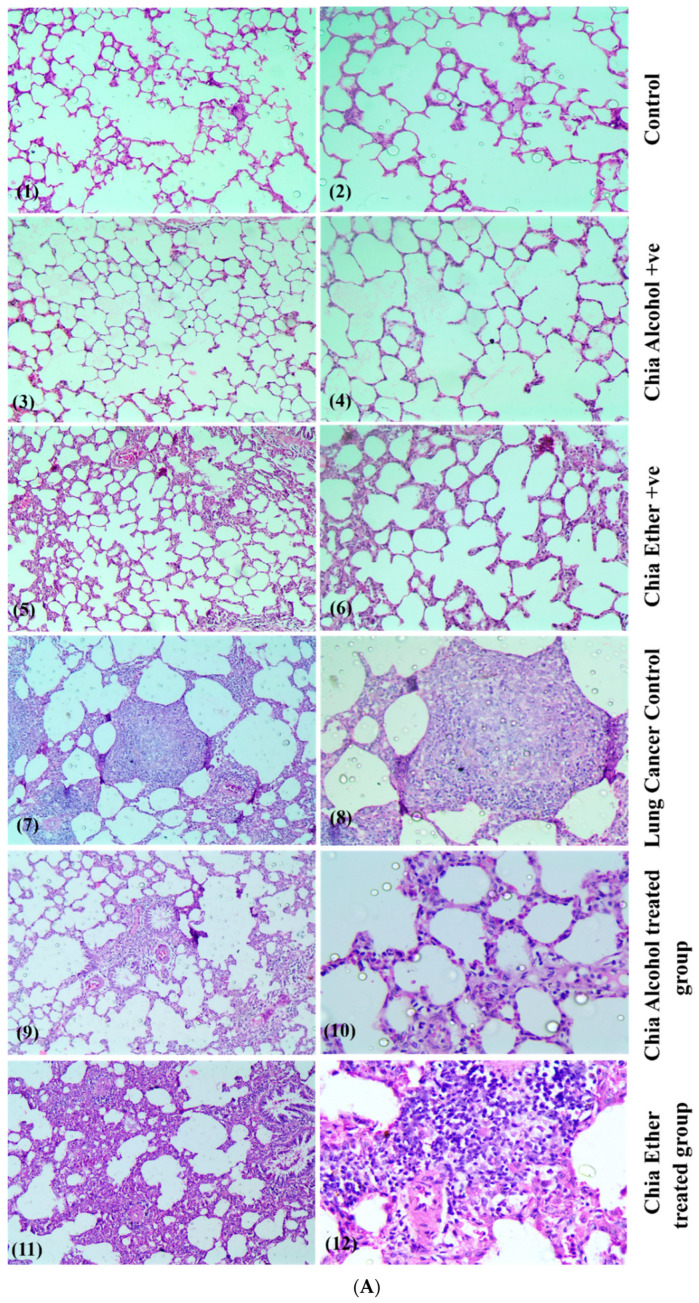
(**A**). Photomicrographs of rat lung sections stained with H&E. −ve control, chia ethanolic extract +ve group, and chia ethanolic extract +ve group showing unremarkable histopathological alterations with normal alveolar architecture and thin interalveolar septa (**1–6, respectively**). (**B**). Photomicrographs of rat testis sections stained with H&E. −ve control, chia ethanolic extract +ve group, and chia ethanolic extract +ve group showing normal seminiferous tubules of regular contour, normal spermatogenesis, and narrow intertubular spaces with the presence of Leydig cells (**1–6, respectively**).

**Figure 5 pharmaceuticals-17-01129-f005:**
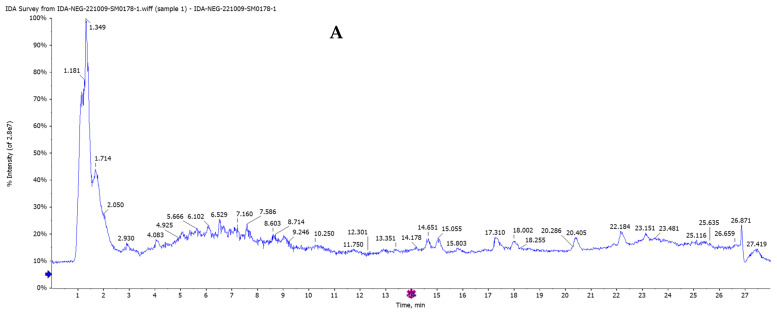
(**A**): Total ion current chromatogram (TIC), and (**B**): Base peak chromatogram of chia ethanolic extract using HPLC-QTOF/HR-MS/MS in (−ve) ionization mode.

**Figure 6 pharmaceuticals-17-01129-f006:**
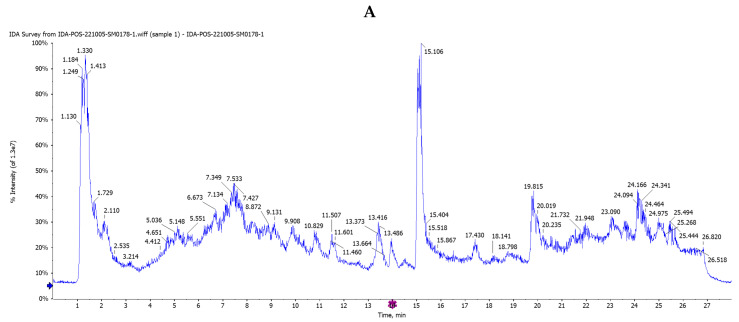
(**A**)**:** Total ion current chromatogram (TIC), and (**B**): Base peak chromatogram of chia ethanolic extract using HPLC-QTOF/HR-MS/MS in (+ve) ionization mode.

**Figure 7 pharmaceuticals-17-01129-f007:**
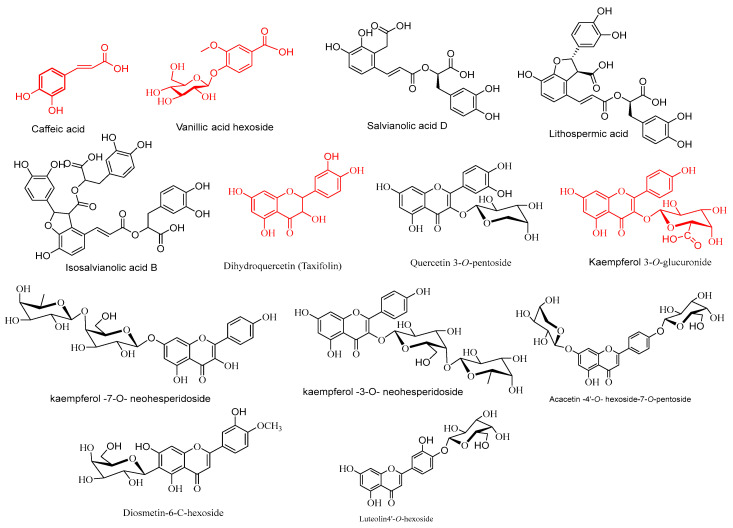
The structure skeleton of the main phenolics and flavonoids tentatively characterized in (−ve) mode from ethanol extract of *Salvia hispanica* seeds by HPLC/HR-QTOF-MS/MS; the most highly concentrated compounds in −ve mode are represented in red.

**Figure 8 pharmaceuticals-17-01129-f008:**
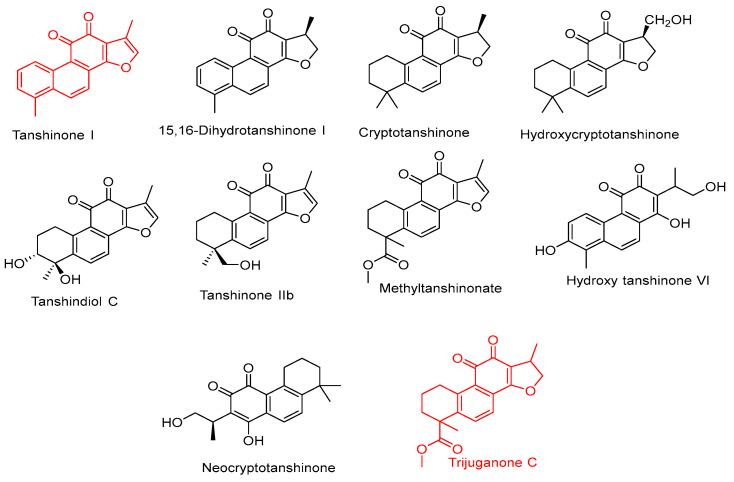
The structure skeleton of the main diterpenoids was tentatively characterized in (+ve) mode from ethanol extract of *Salvia hispanica* seeds by HPLC/HR-QTOF-MS/MS; the most highly concentrated compounds in +ve mode represented in red.

**Figure 9 pharmaceuticals-17-01129-f009:**
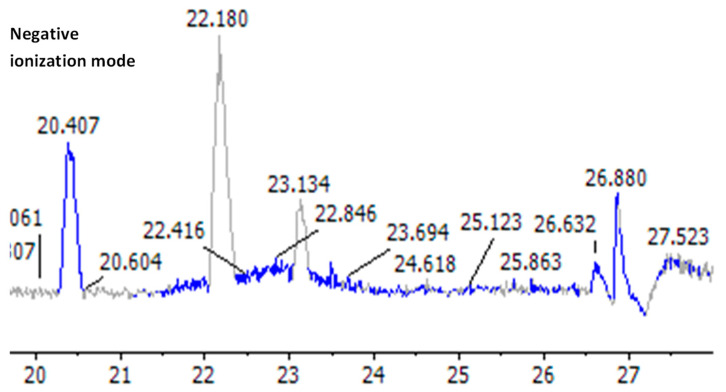
Base peak chromatograms (BPCs) of the fatty acids tentatively identified in petroleum ether extract of *Salvia hispanica* seeds in negative (−ve) mode; Fatty acids are represented in blue.

**Table 1 pharmaceuticals-17-01129-t001:** The effect of the administration of chia ether and alcohol extracts in rats for five months on lung antioxidants.

Groups	GSH (mmm/L)	GR (U/g)	GST (U/g)	GPx (U/g)
Negative control	4.76 ± 0.15 ^d^	3.89 ± 0.24 ^e^	3.02 ± 0.41 ^e^	2.26 ± 0.31 ^c^
NNK-control	3.20 ± 0.11 ^e^	2.80 ± 0.07 ^f^	2.44 ± 0.05 ^f^	1.66 ± 0.04 ^d^
Chia Ether-treated	5.73 ± 0.41 ^b^	4.84 ± 0.38 ^b^	3.95 ± 0.38 ^b^	3.08 ± 0.30 ^a^
Chia Alcohol-treated	4.81 ± 0.14 ^d^	4.14 ± 0.05 ^d^	3.47 ± 0.17 ^d^	2.67 ± 0.13 ^b^
Ether ve+	6.75 ± 0.21 ^c^	5.57 ± 0.17 ^c^	4.40 ± 0.15 ^c^	3.08 ± 0.10 ^a^
Alcoholic ve+	5.25 ± 0.05 ^a^	4.48 ± 0.05 ^a^	3.72 ± 0.11 ^a^	2.90 ± 0.09 ^a^

Each value is the meaning of six replicates. Data were analyzed via the one-way ANOVA test for comparisons among means at *p* ≤ 0.05. Values with the same superscript letter are not significantly different, while values with different letters are significantly different at *p* ≤ 0.05.

**Table 2 pharmaceuticals-17-01129-t002:** Effect of chia ether and ethanolic extracts on liver functions of NNK-induced lung cancer in rats during five months.

Groups	Total Protein (g/dL)	Albumin (g/dL)	Globulin (g/dL)	Albumin/Globulin	GPT (U/L)	GOT (U/L)	GOT/GPT Ratio
Negative control	7.86 ± 0.05 ^a^	3.38 ± 0.02 ^ab^	4.48 ± 0.05 ^a^	0.75 ± 0.01 ^c^	31.08 ± 1.37 ^c^	67.42 ± 2.13 ^c^	2.17 ± 0.03 ^b^
Lung cancer control	5.84 ± 0.20 ^c^	2.54 ± 0.06 ^c^	3.30 ± 0.24 ^d^	0.78 ± 0.07 ^c^	40.93 ± 1.36 ^a^	81.71 ± 1.98 ^a^	2.00 ± 0.02 ^d^
Chia Ether-treated	7.11 ± 0.22 ^b^	3.43 ± 0.14 ^ab^	3.69 ± 0.11 ^c^	0.93 ± 0.03 ^a^	32.94 ± 1.29 ^b^	70.11 ± 1.87 ^b^	2.13 ± 0.03 ^c^
Chia Alcohol-treated	7.14 ± 0.21 ^b^	3.44 ± 0.18 ^ab^	3.70 ± 0.19 ^c^	0.93 ± 0.08 ^a^	30.15 ± 1.09 ^c^	66.07 ± 1.59 ^c^	2.19 ± 0.03 ^ab^
Chia Ether ve+	8.00 ± 0.05 ^a^	3.54 ± 0.02 ^ab^	4.46 ± 0.05 ^a^	0.79 ± 0.01 ^c^	31.08 ± 0.88 ^c^	67.42 ± 1.28 ^c^	2.17 ± 0.02 ^b^
Chia Alcohol ve+	7.76 ± 0.30 ^a^	3.57 ± 0.09 ^a^	4.19 ± 0.28 ^b^	0.86 ± 0.06 ^b^	29.25 ± 1.00 ^c^	64.76 ± 1.45 ^c^	2.21 ± 0.03 ^a^

Each value is the meaning of six replicates. Data were analyzed via the one-way ANOVA test for comparisons among means at *p* ≤ 0.05. Values with the same superscript letter are not significantly different, while values with different letters are significantly different at *p* ≤ 0.05.

**Table 3 pharmaceuticals-17-01129-t003:** Effect of chia ether and ethanolic extracts on renal functions of NNK-induced lung cancer in rats during five months.

Groups	Uric Acid (mg/dL)	Urea (mg/dL)
Negative control	5.65 ± 0.11 ^bc^	54.68 ± 2.11 ^c^
Lung cancer control	7.14 ± 0.16 ^a^	62.77 ± 0.72 ^a^
Chia Ether	5.86 ± 0.11 ^b^	58.76 ± 0.24 ^b^
Chia Alcohol	5.75 ± 0.41 ^bc^	56.08 ± 1.19 ^c^
Chia Ether ve+	5.90 ± 0.11 ^b^	52.01 ± 2.11 ^d^
Chia Alcohol ve+	5.41 ± 0.39 ^c^	54.34 ± 1.16 ^c^

Each value is the meaning of six replicates. Data were analyzed via the one-way ANOVA test for comparisons among means at *p* ≤ 0.05. Values with the same superscript letter are not significantly different, while values with different letters are significantly different at *p* ≤ 0.05.

**Table 4 pharmaceuticals-17-01129-t004:** Metabolites tentatively identified in ethanol extract of *Salvia hispanica* seeds by HPLC/QTOF-MS/MS in both negative (−ve) and positive (+ve) ionization modes.

No.	RT (min)	Name	Class	[M-H]^−^ *m*/*z*	[M+H]^+^ *m*/*z*	Diff (ppm)	MF	MS^2^
Phenolics								
1	0.949	Isocitric acid	Organic acid	191.0198	193.2103	1.19	C_6_H_8_O_7_	173.0078 [M-H-H_2_O]^−^, 111.0085, 87.0089, 85.0298,
2	0.976	Citric acid	Organic acid	191.0213	193.1380	0.42	C_6_H_8_O_7_	173.0085 [M-H-H_2_O]^−^, 129.0204 [M-H-H_2_O-CO_2_]^−^ 85.0295, 87.0089, 175.0100,
3	0.988	Succinic acid	Organic acid	117.0223		0.63	C_4_H_6_O_4_	99.0094, 73.0309
4	1.016	* Quinic acid	Organic acid	191.0200	193.1018	0.11	C_7_H_12_O_6_	173.0100 [M-H-H_2_O]^−^, 146.0619 [M-H-COOH]^−^ 85.0305, 87.0095, 114.0561,
5	1.093	*** Caffeic acid	Hydroxy-cinnamic acids	179.0551	181.1023	2.56	C_9_H_8_O_4_	161.0420 [M-H-H_2_O]^−^, 135.0402 [M-H-CO_2_]^−^, 75.0085, 163.0899, 137.0720
6	1.259	** Ferulic acid	Hydroxy-cinnamic acids	193.717	195.1176	1.70	C_10_H_10_O_4_	177.0700, 175.0609 [M-H-H_2_O]^−^, 157.0389 [M-H-2H_2_O]^−^, 149.0555 [M-H-CO_2_]^−^, 151.0749
7	1.352	** Rosmarinic acid	Hydroxy-cinnamic acids	359.2063	361.0489	0.72	C_18_H_16_O_8_	315.1075 [M-H-CO_2_]^−^, 311.1118, 299.0021, 271.1119, 179.0569 [M-H-(C_9_H_9_O_4_)^•^]^−^,181.0706, 301.1221
8	1.390	** Coumaric acid	Hydroxy-cinnamic acids	163.0413	165.0405	0.32	C_9_H_8_O_3_	119.0504 [M-H-CO_2_]^−^, 147.0315 [M-H-H_2_O]^−^, 149.1924, 121.0315
9	1.583	* Protocatechueic acid	Hydroxy-benzoic acids	153.0200	155.1495	2.71	C_7_H_6_O_4_	135.000 [M-H-H_2_O]^−^, 137.0261, 109.0315 [M-H-CO_2_]^−^, 111.0347, 84.9906
10	1.606	* Przewalskinic acid	Hydroxy-cinnamic acids	356.9589	359.1562	2.10	C_18_H_14_O_8_	313.1408 [M-H-COOH]^−^, 313.9870, 179.1068 [M-H-C_9_H_8_O_4_]^−^, 197.1156, 189.9389, 135.1153
11	1.761	* Danshensu	Hydroxy-cinnamic acids	197.09175	199.1730	1.93	C_9_H_10_O_5_	152.9751 [M-H-CO_2_]^−^, 112.9854, 181.1058 [M-H-H_2_O]^−^
12	2.470	Vanillic acid	Hydroxy-benzoic acids	167.0340	169.0840	0.22	C_8_H_8_O_4_	149.0223 [M-H-H_2_O]^−^, 122.9672 [M-H-CO_2_]^−^, 151.1080, 125.0663, 93.0666
13	2.548	P-hydroxy benzoic acid	Hydroxy-benzoic acids	137.0249	139.1209	0.15	C_7_H_6_O_3_	121.0388 [M-H-H_2_O]^−^, 93.0345, 97.0660,
14	4.064	* Protocatechuic aldehyde		137.0235		0.30	C_7_H_6_O_3_	108.0211 [M-H-CHO]^−^, 109.0284 [M-H-CO]^−^
15	8.0540	D-(+)-Galacturonic acid	Hydroxy-cinnamic acids	193.0469		4.71	C_6_H_10_O_7_	161.0206, 149.0924
16	15.008	Salvianolic acid A	Hydroxy-cinnamic acids	493.2001		6.34	C_26_H_22_O_10_	431.1568 [M-H-H_2_O-CO_2_]^−^, 356.9344 [M-H-2H_2_O]^−^, 294.9048 [M-H-198(C_9_H_9_O_5_^•^]^−^, 248.9587, 288.9440, 256.9281, 384.9308, 364.9154, 325.1803, 256.9281,
17	15.781	* Salvianolic acid F	Hydroxy-cinnamic acids	313.0713		3.62	C_17_H_14_O_6_	294.903, 269.0936 [M-H-CO_2_]^−^,267.2695, 255.2312, 188.9408, 153,0566, 147.0476, 135.9427, 228.9303
18	16.728	* Salvianolic acid L	Hydroxy-cinnamic acids	717.2300		1	C_36_H_30_O_16_	672.8041 [M-H-COOH]^−^, 518.8446 [M-H-198]^−^, 501.5309 [M-H-198-18]^−^, 549.2213, 617.1651, 573.1695, 586.8670
19	17.279	* Isosalvianolic acid B	Hydroxy-cinnamic acids	717.2853		1.83	C_20_H_18_O_10_	671.2583 [M-H-COOH]^−^, 581.2956, 518.8367 [M-H-198]^−^, 481.1706, 345.2053
20	17.898	* Salvianolic acid D	Hydroxy-cinnamic acids	417.2852		0.52	C_20_H_18_O_10_	373.2140 [M-H-CO_2_]^−^, 354.9166 [M-H-H_2_O-CO_2_]^−^, 349.2291, 326.9340, 286.9384, 248.9610
21	20.368	* Lithospermic acid	Hydroxy-cinnamic acids	537.3337		2.36	C_27_H_22_O_12_	492.8111 [M-H-44]^−^, 316.9444, 248.9591, 180.9715, 112.9853
Phenolic acid hexosides								
22	1.044	* Protocatechueic acid-*C*-hexoside	Hydroxy-benzoic acids	315.0049		1.7	C_13_H_16_O_9_	195.0514 [M-H-120]^−^, 225.0023 [M-H-90]^−^ 271.0386 [M-H-CO_2_]^−^, 75.0092
23	1.210	*** Vanillic acid *O*-hexoside	Hydroxy-benzoic acids	329.0383	331.0652	0.75	C_14_H_18_O_9_	285.0926 [M-H-CO_2_]^−^, 167.0361. [M-H-162(hex)]^−^, 123.0461, 122.0390
24	1.288	** Ferulic acid *O*-hexoside	Hydroxy-cinnamic acids	355.0736		1.43	C_16_H_20_O_9_	193.0508 [M-H-162(hex)]^−^, 178.0272, 163.0407, 85.0295
25	1.326	** Proto- catechueic acid-*O*-hexoside	Hydroxy-benzoic acids	315.0720		3.00	C_13_H_16_O_9_	153.0196 [M-H-162(hex)]^−^, 123.0437, 109,0290, 108.0217
26	1.352	** Caffeic acid *O*-hexoside	Hydroxy-cinnamic acids	341.0981	343.1373	1.12	C_15_H_18_O_9_	179.0345 [M-H-162(hex)]^−^, 135.0447 [M-H-162(hex)-CO_2_]^−^, 181.0703, 164.0762 [M-H-caffiec]^−^
27	1.431	* Hydroxy- benzoic acid O-hexoside	Hydroxy-benzoic acids	299.0774	301.1236	3.02	C_13_H_16_O_8_	137.0236 [M-H-162(hex)]^−^, 139,0505 239.1370, 241.1230, 39.0348
28	1.803	Caffeoylquinic acid	Hydroxy-cinnamic acids	353.0816	355.0741	0.22	C_16_H_18_O_9_	191.0497 [M-H-162]^−^ 193.0686, 145.0970
29	1.977	Quinic acid di hexose	Organic acid glycoside	515.1584		0.85	C_19_H_32_O_16_	352.9196 [M-H-162(hex)]^−^, 191.1060 [M-H-324 (di hex)]^−^, 173.0922 [M-H-324 (di hex)-H_2_O]^−^, 139.0749
30	2.049	Caffeic acid di rhamnoside	Hydroxy-cinnamic acids	471.1412		3.54	C_21_H_28_O_12_	427.4572 [M-H-CO_2_]^−^, 324.9018 [M-H-146(rham)]^−^, 145.9292 [M-H-146(rham)-179(Caffoyl]^−^
Flavonoids (flavonols)								
Kaempferol derivatives								
31	1.122	*** Kaempferol-3-*O*-Glucuronide	Flavonol	461.0726		0.99	C_21_H_18_O_12_	284.9790 [M-H-176(gluc)]^−^
32	1.314	* Dihydro kaempferol-3-O-hexoside	Flavonol	449.201		1.87	C_21_H_12_O_11_	431.2058 [M-H-H_2_O]^−^, 287.0200 [M-H-162(gl)]^−^, 269.1382, 251.1277
33	1.978	Kaempferol-3-*O*-α-L-rhamnoside	Flavonol	431.1766		2.12	C_21_H_20_O_10_	285.0432 [M-H-146(rham)]^−^
34	6.475	* kaempferol-3-*O*-neohesperido- side	Flavonol	593.1432		3.66	C_28_H_32_O_15_	447.1335 [M-H-146(rham]^−^, 431.0731 [M-H-162(gl)]^−^, 285.0410, 284.0324
35	6.530	* kaempferol-7-*O*-neohesperdo- side	Flavonol	593.1189	595.1794	1.15	C_28_H_32_O_15_	447.0907 [M-H-146(rham]^−^, 431.1450 [M-H-162(gl)]^−^, 284.0359 [M-H-162(gl)-146(rham]^−^, 285.3020, 433.1600, 287.0339
36	6.698	Kaempferol-3-*O*-hexoside	Flavonol	447.0922	449.2290	0.33	C_21_H_20_O_11_	284.0303 [M-H-162(gl)]^−^, 285.0370, 255.0929, 287.0866
37	7.149	* Kaempferol-3-*O*-α-L- arabinoside	Flavonol	417.0794		0.94	C_20_H_18_O_10_	284.0307, 285.0494 [M-H-132(ara)]^−^
38	8.628	* Kaempferol 3-*O*-pentoside-7-*O*-glucuronide	Flavonol	593.3003		1.50	C_26_H_26_O_16_	461.2058 [M-H-132(pent)], 285.0268 [M-H-132(pent)176(gluc)]^−^
39	8.600	Kaempferol 3,7-*O*-bis-alpha-L-rhamnoside	Flavonol	577.2154		2.84	C_27_H_30_O_14_	430.0999 [M-H-146(rham)]^−^, 285.0310 [M-H-292(2rham)]^−^
40	9.876	Kaempferol 4′-*O*-methoxy	Flavonol	299.0969	301.1364	0	C_16_H_12_O_6_	284.0445 [M-H-CH_3_]^−^, 286.0133, 153.0716
41	14.577	* Kaempferol 3-*O*-glucuronide-7-*O*-pentoside	Flavonol	593.2150		0	C_26_H_26_O_16_	416.8290 [M-H-176(gluc)]^−^, 285.0348 [M-H-176(gluc)-132(pent)]^−^
42	17.0317	Kaempferol 3-*O*-robinoside-7-*O*-rhamnoside	Flavonol	739.3776		3.65	C_33_H_40_O_19_	593.3730 [M-H-146(rham)]^−^, 431.1500 [M-H-162(gl)-146(rham]^−^, 285.0530 [M-H-146(rham)-308]
Quercetin derivatives								
43	6.073	Quercetin-3-*O*- hexoside	Flavonol	463.0866		3.22	C_21_H_20_O_12_	300.0251, 301.0265 [M-H-162(gl)]^−^, 271.0208
44	7.099	* Quercetin-3-*O*-pentoside	Flavonol	433.2067		0.61	C_20_H_18_O_11_	301.1034 [M-H-132(pent)]^−^, 261.1120, 228.9288
45	7.353	* Isorhamentin-3-*O*-hexoside	Flavonol	447.2007		3.10	C_22_H_22_O_12_	314.0445, 315.0410 [M-H-162(gl)]^−^, 301.3301 [M-H-162(gl)-CH_3_]^−^, 355.1398, 285.0468 [M-H-162(gl)-CH_3_-OH]^−^, 294.4905
46	7.392	* Quercetin-7-*O*-hexoside	Flavonol	463.0358		0.28	C_21_H_20_O_12_	301.0333, 300.0265, 285.0301
47	7.573	* Isorhamnetin-3-*O*-rhamnoside-7-*O*-hexoside	Flavonol	623.2701	625.3426	7.30	C_28_H_32_O_16_	577.2863 [M-H-146(rham)]^−^, 461.1439 [M-H-162(gl)]^−^, 317.1066 [M-H-146(rham)-162(gl)]^−^, 463.2891, 256.9902
48	11.414	Isorhamentin-3-*O*-hexoside-7-*O*-pentoside	Flavonol	609.2966		3.62	C_16_H_30_O_16_	446.9234 [M-H-162(gl)]^−^, 314.0422 [M-H-162(gl)-132(pen)]^−^, 301.516 [M-H-162(gl)-132(pen)-CH_3_]^−^
49	14.108	Isorhamentine-3-*O*-pentoside-7-*O*-glucuronide	Flavonol	623.2062		4.44	C_27_H_28_O_17_	491.2102 [M-H-132(pent)]^−^,447.2551 [M-H-176]^−^
50	14.168	Quercetin-3-*O*- rhamnoside-7-*O*-glucuronide	Flavonol	623.2340		1.61	C_27_H_28_O_17_	477.2551 [M-H-146(rham)]^−^, 447.1668 [M-H-176(glu)]^−^, 301.0022
51	14.438	* 3′,4′-di methyl quercetin	Flavonol	329.1042		1.49	C_17_H_14_O_7_	314.0796 [M-H-CH_3_]^−^, 299.0562 [M-H-2CH_3_]^−^, 283.2662, 255.0794
52	14.629	* Isorhamnetin 3-*O*-pentoside-7-*O*- rhamnoside	Flavonol	593.2672		1.25	C_27_H_30_O_15_	460.9255 [M-H-132(pent)]^−^,447.1633 [M-H-146(rham)]^−^, 299.5170 [M-H-132(pent)-146(rham)-CH_3_]^−^, 284.5016
53	15.105	Axillarin	Flavonol	345.0562		0.55	C_17_H_14_O_8_	327.1058 [M-H-H_2_O]^−^, 315.0480 [M-H-2CH_3_]^−^, 301.1271, 299.0208 [M-H-OCH_3_-CH_3_^]−^, 283.1136 [M-H-2OCH_3_]^−^, 329.1173, 311.0982, 241.0972, 329.1811, 312.1949, 285.2878
Myricetin derivatives								
54	1.379	*** Myricitrin	Flavonol		463.1352	0.91	C_21_H_20_O_12_	317.0900 [M-H-146(rham)]^+^, 283.1069
55	4.241	Syringetin-3-*O*-hexoside	Flavonol	507.1362	509.1349	0.53	C_23_H_24_O_13_	461.1482 [M-H-CH_3_-H_2_O]^−^, 345.1182 [M-H-162(gl)]^−^, 347.5120, 435.0228, 286.9434, 294.8986, 400.8672, 283.1105, 218.9472, 327.1020, 315.1098
56	5.811	Syringetin-3-*O*-hexoside-7-*O*- glucuronide	Flavonol	683.2351		0.12	C_29_H_32_O_19_	521.1075 [M-H-gl (162)]^−^, 345.1395 [M-H-gl (162)-gluc(176)]^−^, 314.8951
Flavonoids (flavones)								
57	4.040	Diosmetin 7-*O*-pentoside	Flavone	431.1327		0.81	C_21_H_20_O_10_	301.1066, 299.0884 [M-H-132(pent)]^−^, 267.9964 [M-H-132(pent)-OCH_3_]^−^
58	5.048	Apigenin7-*O*- hexoside	Flavone	431.1439	433.1628	0.13	C_21_H_20_O_10_	269.1038 [M-H-162(gl)]^−^, 271.1914, 283.1165, 227.1060
59	4.102	Diosmetin 8-*C*-hexoside-7-*O*- pentoside	Flavone	593.2098		−5.2	C_27_H_30_O_15_	473.1159 [M-H-120]^−^, 431.1121 [M-H-120-132(pent]^−^, 298.0425, 284.4091
60	5.550	Diosmetin 6-*C*-pentoside	Flavone		433.1143	0.67	C_21_H_20_O_10_	373.1115 [M-H-60]^−^, 343.1513, [M-H-90]^−^ 301.1092, 271.1412
61	5.930	Diosmetin 5-*O*-robinoside-7-*O*-pentoside	Flavone	739.2645		0.96	C_33_H_40_O_19_	607.2067 [M-H-132(pent)]^−^, 577.2339 [M-H-162(gl)]^−^, 298.9051 [M-H-132-308]^−^
62	6.052	Vitexin-7-*O*-hexoside	Flavone	593.2286		3.15	C_27_H_30_O_1_	473.2267 [M-H-120]^−^, 269.1150 [M-H-324(digl)]^−^
63	6.422	Acacetin di-*O*-hexoside	Flavone	607.2552		4.33	C_28_H_32_O_15_	444.9950 [M-H-162(pent)]^−^, 282.9815
64	6.638	* Acacetin-4′-*O*-hexoside-7-*O*- pentoside	Flavone	577.1778	579.2121	0.49	C_27_H_30_O_14_	415.1565 [M-H-162(gl)]^−^, 282.9325 [M-H-162(gl)-pent(132)]^−^, 179.0547, 417.1486, 285.2756, 342.0983
65	6.940	* Luteolin-4′-*O*- hexoside	Flavone	447.2077	449.1687	0.80	C_21_H_20_O_11_	287.0590 [M+H-162(gl)]^+^, 270.1509, 262.0928, 269.1466, 284.0303, 285.0370 [M-H-162(gl)]^−^, 334.8963, 255.0292
66	7.274	* Diosmetin-7-*O*-hexoside	Flavone	461.0213		1.89	C_22_H_22_O_11_	430.4532 [M-H-OCH_3_)]^−^, 299.0960 [M-H-162(gl)]^−^, 276.7211 [M-H-162(gl)-OCH_3_]^−^
67	7.642	* Luteolin-7-*O*- hexoside	Flavone	447.0972	449.0168	2.51	C_15_H_12_O_4_	285.0461 [M-H-162(gl)]^−^, 284.0372, 355.1246, 334.8923, 227.0360, 256.9271, 243.0273, 287.0590, 270.1509, 251.1326
68	7.495	* Diosmetin-6-*C*-hexoside	Flavone	461.1483	463.1213	0.76	C_22_H_22_O_11_	341.0640 [M-H-120]^−^, 370.8851 [M-H-90]^−^, 299.0790, 286.8904, 301.1072, 343.1250, 283.1101
69	7.788	* Pinocembrin	Flavone	255.2329		0.28	C_15_H_12_O_4_	237.0635, 210.9630, 155.0319
70	8.690	* 5,7, 3′-Trihydroxy-6, 4′, 5′-trimethoxy flavone	Flavone	359.2780	361.1542	2	C_81_H_16_O_8_	344.421 [M-H-CH_3_]^−^, 329.1391 [M-H-2CH_3_]^−^, 314.1921 [M-H-3CH_3_]^−^, 331.1248, 315.1230, 343.1713
71	10.892	Baicalein-7-*O*- glucuronide	Flavone	445.3198		0	C_21_H_18_O_11_	401.2819 [M-H-COO]^−^, 399 [M-H-COOH]^−^, 269.1280 [M-H-176(gluc)]^−^
72	11.041	Sorbifolin	Flavone		301.1468	0.40	C_16_H_12_O_6_	286.0480 [M-H-CH_3_]^−^, 268.0123 [M-H-CH_3_-H_2_O]^−^, 283.1083, 255.0924
73	14.549	* Diosmetin-5-*O*-rhamnoside-7-*O*-pentoside	Flavone	577.2592		3.57	C_27_H_30_O_14_	431.7884 [M-H-132(pent)], 445.0339 [M-H-146(rham)]^−^, 299.0445 [M-H-132(pent)-146(rham)]^−^
74	17.010	3,5, 7-trihydroxy-4′-methoxy-flavone (diosmetin)	Flavone	299.2039	301.1468	4.18	C_16_H_12_O_6_	284.0333 [M-H-CH_3_]^−^, 271.0174, 256.0266, 153.0532, 117.0362, 286.0480, 301.1468, 286.0480
Flavanones								
75	8.263	Isosakuranetin-7-*O*-neohesperido-side	Flavanone	593.1828		1	C_28_H_34_O_14_	447.0959, 285.2212
76	4.930	Daidzein-8-*C*- hexoside (Puerarin)	Isoflavonoid C-glycosides	415.0411	417.1523	0.83	C_21_H_20_O_9_	352.8659, 327.1397, 303.1324, 329.1250, 253.1495, 237.0435, 255.1210
Flavan-3-ol								
77	1.458	(+)-Catechin	Flavan-3-ol	289.1634	291.0704	1.7	C_15_H_14_O_6_	271.1141 [M-H-H_2_O]^−^, 243.0639, 245.0942 [M-H-CO_2_]^−^, 203.0835, 153.315, 273.0993
78	1.363	(+)-Gallocatechin	Flavan-3-ol	305.0738		7.5	C_15_H_14_O_7_	179.0606 [M-H-125(C_6_H_5_O_3_^•^)]^−^, 261.1294, 287.0953, 225.1206
Tannins								
79	5.976	Ellagic acid-pentoside		433.2188	435.1711	−3.6	C_19_H_14_O_12_	301.0137 [M-H-132(pent)]^−^, 256.9320, 228.9299, 418.1720, 372.1430, 347.2093, 238.1145
80	4.226	Ellagic acid-hexoside		463.6302		−5.1	C_20_H_16_O_13_	299.9887, 258.9403, 256.9124
Anthocyanins								
81	6.237	Delphinidin-3-*O*-glucopyranoside	Anthocyanin	463.0816		4.24	C_21_H_21_O^+^_12_	300.276, 301.0343 [M-H-162(gl)]^−^, 294.9091, 271.0208
82	8.700	Delphinidin-3-*O*-(6″-*O*-*α*-rhamno-pyranosyl-*β*- glucopyranosid)	Anthocyanin	609.3502		2.65	C_27_H_31_O^+^_16_	563.3040 [M-H-146(rham)]^−^, 447.1565 [M-H-162(gl)]^−^
83	7.376	* Phloretin	Dihydro chalcones	272.9206		5.21	C_15_H_14_O_5_	228.9333, 188.9393
84	9.953	Phloretin 2′-glucoside	Phloretin-hexoside (Phlorizin)	435.1531		0.36	C_21_H_24_O_10_	372.8508, 310.8799, 374.8417, 273.1094
Stilbenes								
85	8.163	E-3,4,5′-Trihydroxy-3′-glucopyranosylstilbene	Stilbene	405.1245		0.12	C_20_H_22_O_9_	336.8967 [M-H-4OH]^−^, 296.9058 [M-H-C_6_H_5_O_2_•]^−^, 272.0836 [M-H-C_8_H_7_O_2_•]^−^, 243.1549 [M-H-162(gl)]^−^
86	13.580	* Pinoresinol	*Tetrahydro* *furan lignan*		359.2283	0.57	C_20_H_22_O_6_	315.12159810 [M+H-CH_3-_H_2_O]^+^, 329.00109810 [M+H-2CH_3_]^+^, 3132013. 155.0931, 297.11529810 [M+H-2OCH_3_]^+^
Aglycones								
87	1.160	*** Dihydro-quercetin (taxifolin)	Flavonol	303.1398	305.1232	0.29	C_15_H_12_O_7_	285.1118 [M-H-H_2_O]^−^, 267.0891 [M-H-2H_2_O]^−^, 243.0154, 287.1253, 257.0932
88	6.327	Isorhamentin	Flavonol	315.0540	317.1157	0.54	C_16_H_12_O_7_	300.0216 [M-H-CH_3_)]^−^, 284.0247, 255.0316, 302.0400, 299.9887, 269.2421, 285.0205 [M-H-OCH_3_)]^−^, 287.0205, 271.0941, 257.0857
89	7.181	* Naringenin	Flavone	270.9701	273.1741	0.43	C_15_H_12_O_5_	225.1045, 255.1765, 159.0904, 129.0704
90	8.655	* Querecetin	Flavonol	301.0004	303.1149	0.93	C_15_H_10_O_7_	283.9982, 255.2302, 229.0132, 151.0066, 285.1116,257.1038, 288.0892, 177.1100
91	9.017	Kaempferol	Flavonol	285.0368	287.1367	0.06	C_15_H_11_O_6_	241.0761, 211.0494, 197.0584, 271.1000, 244.0265, 231.0230, 170.0761, 145.0853
92	9.061	Hesperetin		301.0345	303.1613	0.28	C_16_H_14_O_6_	283.0573, 285.1972, 255.2383, 245.0498, 179.0347, 164.0099, 151.0041, 191.1036,
93	14.440	* Apigenin (Genistein)	Flavone	269.2099	271.3207	0.53	C_15_H_10_O_5_	254.1110, 251.2067, 225.2285, 134.0233, 253.54302, 229.2165,
94	15.088	*** Acacetin (Linarigenin)	Flavone	283.0626		0.06	C_16_H_12_O_5_	268.0371 [M-H-CH_3_)]^−^, 240.0426, 239.0331, 225.0188
95	15.025	** Myricetin	Flavonol	317.0236	319.3023	0.77	C_15_H_10_O_8_	301.6302, 283.2832, 302.2432, 272.9571, 248.9191
96	15.526	Luteolin	Flavone	285.0395		0.72	C_15_H_10_O_6_	271.0354, 257.0208, 241.0452, 229.0522, 211.0665
Coumarins								
97	2.776	Esculin	Coumarin glycoside	339.0652		0.93	C_15_H_16_O_9_	177.0174 [M-H-162(gl))]^−^,
98	6.748	* (Daphnetin)7, 8-dihydroxy coumarins	Dihydroxy coumarins	177.0187	179.1027	6.43	C_9_H_6_O_4_	133.0271, 149.0247 [M-H-CO]^−^ 105.0363 [M-H-_2_H_2_O-CO]^−^, 91.0500, 123.0975
Alkaloids								
99	13.579	* Tembetarine	Alkaloid		345.1332	0.19	C_20_H_26_ NO_4_	327.1227 [M-H-H_2_O]^−^, 297.0772 [M-H-2CH_3_-H_2_O]^−^, 209.0451, 107.0483
100	15.398	* Jatrorrhizine	Alkaloid	337.2053	339.1709	0.62	C_20_H_20_ NO_4_	309.1431 [M-H-OCH_3_]^−^, 293.2282 [M-H-OCH_3_-CH_3_]^−^ 291.2018, 275.2087 [M-H-3CH_3_-H_2_O]^−^, 274.9289 [M-H-2OCH_3_]^−^, 279.2315, 321.2729, 276. 9100 [M-H-2OCH_3_]^−^
101	15.413	* Menisperine	Alkaloid		357.1111	1.70	C_21_H_26_ NO_4_	311.0995 [M-H-OCH_3_-CH_3_]^−^
Diterpene								
102	8.357	6-Hydroxy-7- Methoxy- tremetone	Member of benzofurans		249.1230	1.27	C_14_H_16_O_4_	234.0590 [M+H-CH_3_]+, 231.1038 [M+H-H_2_O]^+^, 187.0729 [M+H-2OCH_3_]^+^, 141.0696 [M+H-C_7_H_8_O^2−^]^+^
103	13.963	Tanshinone IIb	Diterpene		311.0561	1.51	C_19_H_18_O_4_	293.2092 [M + H-H_2_O]^+^, 283.1309 [M + H-CO]^+^, 268.388 [M + H-CO-CH_3_]^+^
104	15.108	*** Trijuganone C	Diterpene		341.0988	0.14	C_20_H_20_O_5_	309.0861 [M+H-32]^+^, 281.5246 [M+H-32-18]^+^, 295.0876 [M+H-COOH]^+^, 311.1120 [M+H-2CH_3_]^+^, 282.1352 [M+H-COOCH_3_]^+^
105	15.169	*** Tanshinone I	Diterpene		277.1856	0.40	C_18_H_12_O_3_	259.9341 [M+H-CH_3_]^+^, 246.9221 [M+H-2CH_3_]^+^, 137.0590 [M-H-C_11_H_82_^•^]^−^, 161.1012 [M+H-CH_3_-C_9_H_82_^•^]^+^
106	15.206	* Tanshinone IIA	Diterpene		295.0962	0.91	C_19_H_18_O_3_	277.1797 [M+H-H_2_O]^+^, 249.1854 [M + H-H_2_O-CO]^+^, 231.1707, [M+H-H_2_O-CO-H_2_O]^+^, 161.0965, 137.0587
107	15.238	Tanshindiol C	Diterpene		313.1525	0.84	C_18_H_16_O_5_	295.0780 [M + H-H_2_O]+, 267.2577 [M + H-H_2_O-CO]+
108	14.743	Hydroxycrypto-tanshinone	Diterpene		313.1068	0.51	C_19_H_20_O_4_	295.0947 [M+H-H_2_O]^+^, 281.1056 [M+H-CH_2_OH]+ , 277.2162
109	15.324	* Neocrypto- tanshinone	Diterpene		315.1334	0.14	C_19_H_22_O_4_	297.1187 [M+H-H_2_O]^+^, 279 [M+H-2H_2_O]^+^, 255.1144 [M+H-C_3_H_7_O^•^]^+^, 163.1406
110	15.740	Methyl- tanshinonate	Diterpene		339.1617	0.27	C_20_H_18_O_5_	307.1825 [M + H-32]^+^, 289.0550 [M + H-32-H_2_O]^+^
111	19.678	* Carnosol	Diterpene		331.2072	0.32	C_20_H_26_O_4_	316.5390 [M+H-CH_3_]^+^, 313.2737, 301.0655 [M+H-2CH_3_]^+^, 271.1711 [M+H-4CH_3_]^+^, 253.1120 [M+H-4CH_3_-18]^+^,
112	19.841	* β-Caryophyllene	*Sesqui-terpenes*		204.1768	0.36	C_15_H_24_	175.6093, 149.0242, 109.952, 93.0347
113	19.905	Crypto- tanshinone	Diterpene		297.1134	1.55	C_19_H_20_O_3_	279.2280 [M + H-18]^+^, 251.2321 [M + H-18-28]^+^, 149.226
114	20.021	* Miltirone	Abietane-type diterpene quinine		283.1572	0.34	C_19_H_22_O_2_	268.3310 [M+H-CH_3_]^+^, 240 [M + H-CH_3_-CO]^+^
115	23.697	15,16-Di-hydro tanshinoneI	Diterpene		279.2340	0.72	C_18_H_14_O_3_	264.0875 [M+H-CH_3_]^+^, 170.9375, 233.5410
116	23.598	Salviacoccin	Diterpene		357.2975	3.35	C_20_H_20_O_6_	339.2889, 290.5238 [M+H-C_4_H_3_O^•^]^+^, 265.2520, 181.1620
117	24.132	Hydroxy-tanshinone VI	Diterpene		313.2731	0.12	C_18_H_16_O_5_	297.9810 [M+H-CH_3_]^+^, 254.4611 [M+H-C_3_H_7_O^•^]^+^, 149.0223, 95.0850 [M+H-C_12_H_11_O_4_^•^]^+^
Amino acids								
118	22.809	* Norvaline	L-alpha-amino acids	115.9319		−1.00	C_5_H_11_NO_2_	72.0441 [M-H-COOH]^−^, 99.0255 [M-H-NH_2_]^−^
Organic compounds								
119	7.919	P-Nitrophenol		138.0231		0.20	C_6_H_5_NO_3_	Not fragments
120	8.630	N-*trans*-Feruloyl-tyramine			314.1380	1.34	C_18_H_19_NO_4_	177.0539, 145.0279, 121.0649

(*–***) Represents the gradation of compound concentrations from lowest to highest (***); the most highly concentrated compounds.

**Table 5 pharmaceuticals-17-01129-t005:** Petroleum ether composition of chia seeds using GC-Mass; Fatty acids and Unsaponifiable matter.

**Fatty Acids**		
**R_t_**	**%Rel.**	**Identified Fatty Acid**
28.62	16.95	Palmitic acid, methyl ester
30.44	2.56	Palmitic acid
32.81	67.48	Linolenic acid, methyl ester
33.42	10.47	Stearic acid, methyl ester
37.20	0.51	11-Eicosenoic acid, methyl ester
37.82	1.27	Eicosanoic acid, methyl ester
**Unsaponifiable matter**		
**R_t_**	**%Rel.**	**Identified Compounds**
4.12	6.59	n-Decane
6.70	0.53	n-Dodecane
11.36	6.44	n-Pentadecane
12.52	0.19	n-Hexadecane
15.14	0.33	n-Octadecane
18.08	0.25	n-Heneicosane
18.97	0.45	n-Docosane
20.79	0.26	n-Tricosane
21.83	0.54	n-Tetracosane
22.74	0.34	n-Pentacosane
23.65	0.24	n-Hexacosane
24.55	1.28	n-Heptacosane
27.87	3.87	Cholesterol
29.35	6.76	Stigmasterol
30.66	43.10	B-Sitosterol
31.49	7.05	B-Sitosterol
32.40	11.32	alpha-amyrine

From the above results, we found that chia seeds are rich in linoleic acid, which represents about 67%. Decane (which makes up roughly 6.6% of the lipid fraction) and n-Pentadecane (which makes up roughly 6.5% of the lipid fraction) are the primary hydrocarbon constituents detected in chia seed lipid fractions. The predominant cholesterol constituent is around 49% β-sitosterole. Additionally, there is 6.8% stigmasterol.

**Table 6 pharmaceuticals-17-01129-t006:** Fatty acids detected in petroleum ether extract of *Salvia hispanica* seeds by HPLC-HR/QTOF-MS/MS in both negative (−ve) and positive (+ve) ionization modes.

No.	RT (min)	Name	Class	[M-H]^−^ *m*/*z*	[M+H]^+^ *m*/*z*	Diff (ppm)	MF	MS^2^
Fatty acids								
1	13.028	* L-Arginine	Amino acid	173.0814		3.8	C_6_H_14_N_4_O_2_	155.0105 [M-H-H_2_O]^−^, 129.0188 [M-H-CO_2_], 111.0097, 85.0310
2	14.041	Glutaric acid	Fatty acid	133.0138		7.3	C_5_H_8_O_4_	115.030 [M-H-CO_2_]^−^, 87.0078 [M-H-H_2_O]^−^
3	15.136	D-3-Phenyllactic acid	Fatty acid		167.1143	11.8	C_9_H_10_O_3_	149.0187 [M-H-H_2_O]^−^, 122.10040 [M-H-COOH]^−^
4	16.028	Suberic acid	di-carboxylic acid		173.0747	6.1	C_8_H_14_O_4_	154.9605 [M-H-H_2_O]^−^, 145.0279 [M-H-CO]^−^, 128.0585 [M-H-COOH]^−^
5	20.079	Citraconic acid	Unsaturated di-carboxylic acid	128.0354	130.0383	4.3	C_5_H_6_O_4_	84.0460 [M-H-CO_2_]^−^
6	21.593	Octadecenoic acid	Fatty acid	281.2466		−5.2	C_18_H_33_O_2_	266.0416 [M-H-CH_3_]^−^, 237.18785 [M-H-CO_2_]^−^, 212.9265, 172.9433
7	22.492	*** Stearic acid	Fatty acid	283.2637		0.3	C_18_H_35_O_2_	239.246 [M-H-CO_2_]^−^, 215.170, 146.9090
8	23.061	Dihydroxy-octadecenoic acid	Fatty acid	313.2400		−2.4	C_18_H_33_O_4_	298.447, 295.2256 [M-H-H_2_O]^−^, 283.0374, 267.2174, 183.0006
9	24.116	Eicosadienoic acid	Fatty acid	307.2700		−0.4	C_20_H_35_O_2_	262.1038 [M-H-COOH]^−^, 238.9305, 170.9435, 112.9843
10	24.572	Hydroxy-octadecadienoic acid	Fatty acid	295.2616		−1.5	C_18_H_31_O_3_	277.043 1 [M-H-H_2_O]^−^, 195.1195, 183.1019, 230.9773, 165.0840
11	25.696	Hydroxy-octadecatrienoic acid	Fatty acid	293.1663		−0.8	C_18_H_29_O_3_	275.1851 [M-H-H_2_O]^−^, 236.1109, 221.15125, 220.1454, 193.1615, 177.0985, 183.0152, 162.0521
12	25.891	Dihydroxy-octadecadienoic acid	Fatty acid	311.1680		−6.5	C_18_H_32_O_4_	293.221 [M-H-H_2_O]^−^, 275.000 [M-H-2H_2_O]^−^, 264.457, 239.0827
13	26.207	Octadecatrienoic acid	Fatty acid	277.2132		0.1	C_18_H_29_O_2_	233.1572, 205.1610, 145.0773
14	26.320	Hydroxy-octadecenoic acid	Fatty acid	297.2771		0.8	C_18_H_33_O_3_	278.9026 [M-H-H_2_O]^−^, 183.0129, 92.9278
15	26.671	** Hydroxy-oxohexadecanoic acid	Fatty acid	285.2703		−3.2	C_16_H_29_O_4_	Not fragment
16	26.720	** Palmitic acid	Fatty acid	255.2306		−1.6	C_16_H_31_O_2_	237.108 [M-H-H_2_O]^−^, 211.0392 [M-H-CO_2_]^−^, 186.9306
17	26.822	** Eicosaenoic acid	Fatty acid	309.1000		−0.2	C_20_H_37_O_2_	265.1532 [M-H-CO_2_]^−^, 247.1215, 240.9238, 172.9363, 104.9527
18	27.515	Octadecadienoic acid	Fatty acid	278.9182		−7.1	C_18_H_31_O_2_	261.0170 [M-H-H_2_O]^−^, 234.9942 [M-H-CO_2_]^−^, 220.1481, 210.9172

(*–***) Represents the gradation of compound concentrations from lowest to highest (***); the most highly concentrated compounds.

**Table 7 pharmaceuticals-17-01129-t007:** Primers used in QRT-PCR.

Gene	Forward Primer (5′-3′)	Reverse Primer (5′-3′)
β-actin	CACGTGGGCCGCTCTAGGCACCAA	CTCTTTGATGTCACGCACGATTTC
c-MYC	CTGTCCATTCAAGCAGACGA	TCCAGCTCCTCCTCGAGTTA
MMP9	CGGCACGCCTTGGTGTAGCA	AGGCAGAGTAGGAGCGGCCC
BCL2	CTC AGTCATCCACAGGGCGA	AGAGGGGCTACGAGTGGGAT

## Data Availability

Samples of the compounds are not available from the authors.

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
