# Peer review of "Chia Seed (Salvia hispanica) Attenuates Chemically Induced Lung Carcinomas in Rats through Suppression of Proliferation and Angiogenesis"

_pharmaceuticals, 2024, doi:10.3390/ph17091129_

Round 1

Reviewer 1 Report

Comments and Suggestions for Authors

This manuscript should significantly be addressed some major requirements below before further consideration:

- The study was the first to report the identification of chemical profiles and some chemical content of chia extracts using LC-MS. This is the major novelty point of this work. However, the novelty concerning the bio-functions of chia extracts must also be claimed. 

- The relationship between bioactivities and major compounds in chia extracts needed to be conducted and discussed in this work. The author may use ACP analysis or virtual study (Docking study) to confirm this issue.

- Furthermore, GCMS data also needs to be added for the full chemical investigation of chia extracts. 

Comments on the Quality of English Language

Minor editing of English language required

Author Response

Author’s Response

Responses to reviewers’ comments for “The therapeutic impact of chia seed (Salvia hispanica) extracts on tumor growth regression in 4-(Methylnitrosamino)-1-(3-pyridyl)-1-butanone (NNK)- induced lung cancer in rats” (Manuscript ID: pharmaceuticals-3127019.).

Dear Editor,

We thank the reviewers for their insightful comments, which enabled us to improve the manuscript. We hope that you will accept it after revision. According to the reviewers' suggestions, the manuscript has been revised carefully. The detailed corrections raised by the reviewers have been addressed point by point and highlighted in yellow in the revised manuscript.

Comments and Suggestions for Authors

This manuscript should significantly be addressed some major requirements below before further consideration:

Commented 1: The study was the first to report the identification of chemical profiles and some chemical content of chia extracts using LC-MS. This is the major novelty point of this work. However, the novelty concerning the bio-functions of chia extracts must also be claimed. 

Response 1: We thank the reviewer for his comments, which enabled us to improve and Polish our work.

  • The study was the first to report the identification of chemical profiles and some chemical content of chia extracts using LC-MS.
  • Also, this study is the first to examine the anticancer effect of chia seed extracts, petroleum ether, and 70% ethanol on lung cancer on an in vivo scale. It is also the first study to use this model: 4-(Methylnitrosamino)-1-(3-pyridyl)-1-butanone (NNK)—induced lung cancer in rats. In addition, we studied how shea extracts work in treating lung cancer.
  • Previous studies have shown the anti-cancer effect of chia oil in lung cancer, but on an in vitro scale using A59 cell lines.
  • The statement was written on page 3, lines 113-118, with yellow highlighting.

Commented 2: The relationship between bioactivities and major compounds in chia extracts needed to be conducted and discussed in this work. The author may use ACP analysis or virtual study (Docking study) to confirm this issue.

Response 2: Actually, we discussed the relationship between demonstrated lung cancer bioactivities and six major compounds in chia extracts (caffeic acid, vanillic acid hexoside, kaempferol-3-O-glucuronide, taxifolin, and two diterpenes: trijugunone C and tanshinon) in the discussion section, page 35-36, lines 789-852.

The anticancer roles of these compounds—caffeic acid, vanillic acid hexoside, kaempferol-3-O-glucuronide, taxifolin, and two diterpenes, trijugunone C and tanshinon—have been confirmed. 

Commented 3: Furthermore, GC-MS data also needs to be added for the full chemical investigation of chia extracts. 

Response 2: The petroleum ether extract of chia was analyzed by GC-MS as the reviewer requested.

  • Materials and methods of GC-MS are on pages 42-43, lines 1009-1052.
  • The results of GC-MS are on pages 29-30, lines 570-581.

Thank you for your attention and help!

Yours Sincerely,
Samah Ali El-Newary
Associated professor at Aromatic and Medicinal Plants Department, National Research Centre,

El-Tahrir St., Dokki, Giza, 12311, Egypt.

Mobile: 01000464073

Reviewer 2 Report

Comments and Suggestions for Authors

Dear Authors

The following points need to be adressed. 

·       Title: The therapeutic word is not suitable in the title as establishing a molecule or formula for therapeutic purposes requires rigorous studies.

·       Abstract: Add a conclusive statement or outcome of the study.

·       Introduction: Add recent statistics on the incidence and mortality rates of lung cancer globally to highlight the significance of the disease.

·       Results: There is no need for the first paragraph in the results section; it is already given in the methods section.

  • Weight and lung histopathology are not biomarkers. Correct this.
  • The second paragraph seems to be the summary of the work, which is not required in the results section.

·       Table 2: Improve the caption to clearly describe the table content and the meaning of superscripts. For example: "Values with the same superscript letter within a column are not significantly different (P ≤ 0.05)."

·       Section 2.2.8: Histology is primarily used to diagnose and characterize lung cancer, not as a biomarker per se. Please correct this.

·       Section 2.3: Herbal medicines and derivative products are widely employed as therapeutics in many nations. Please revise as "Herbal medicines and their derivative products are widely used as therapeutic agents in many countries."

·       The statement "Various mass spectrometric techniques have gradually been used to investigate medicinal plants and profile their secondary metabolites. Chromatography has recently embraced high-performance liquid chromatography/quadrupole time-of-flight mass spectrometry (HPLC/QTOF-MS) technology. Its excellent resolution, rapidity, and sensitivity make it an ideal separation technique" is mostly accurate but can be refined for clarity and precision.

·       Delete this due to repetition: “The initial motivation for this work is to use Salvia hispanica (Chia) seeds belonging to the mint family (Lamiaceae) to identify a viable cure for a dangerous lung cancer disease. Among our other objectives in this study is to investigate the chemical profile of the most active extracts.”

·       This is not the right claim for the study: “Therefore, our goal was to identify the most effective anti-cancer lung compounds.” Rectify this.

  • Font size throughout the manuscript needs to be uniform.

·       Check Table 4 and correct; I think it is diterpene, not “Diterbene.”

·       Alkaloid Section: The alkaloid section needs to be revisited. The abstract and discussion inform that one alkaloid is present, but in the alkaloid section, the number mentioned is 04, and the names assigned are 03. Additionally, it should be noted that adenosine is not an alkaloid. The claim that chia seeds contain alkaloids would require substantial scientific evidence from phytochemical analysis studies. If such evidence is not available, the assumption should be that chia seeds do not contain notable levels of alkaloids.

  • What reference compounds were used in the LCMS analysis?

·       Figure 6: The need for optimization to achieve better resolution and accurate identification of compounds. 1.136 - 1.322 minutes: Early peaks showing potential overlap. 13.435 - 15.324 minutes: Several closely eluting peaks in the middle region. 19.750 - 20.209 minutes: Late peaks with close elution times. The provided chromatogram may not be acceptable for publication in its current state due to insufficient resolution of several peaks. The lack of clear separation between closely eluting compounds can compromise the specificity, accuracy, and quantitative reliability of the analysis. To meet the publication standards, the chromatographic conditions should be optimized to achieve better separation and resolution of the peaks. This could involve adjusting the gradient program, changing the column, altering the flow rate, or modifying the mobile phase composition.

·       Figure 7: The separation in the positive ionization mode is not fully acceptable for publication due to insufficient resolution of several peaks.

·       Compound Classification: L-Arginine is not a fatty acid. It is an amino acid.

  • Suberic acid, also known as octanedioic acid, is a dicarboxylic acid. It is not a fatty acid.
  • Citraconic acid, also known as methylmaleic acid, is an unsaturated dicarboxylic acid. It is not a fatty acid. It will be better to recheck the classification
  • The grouping of certain compounds needs to be revised or grouped under miscellaneous if they do not fit a particular class of compounds.

·       Diff (ppm) Values: In Table 4 & 5, Diff (ppm) values for up to 7 peaks are higher than 5 ppm, indicating poor precision. A value for anthocyanins is 42. A deviation of 42 ppm indicates that the measured m/z value is significantly different from the expected value, which could lead to incorrect identification of the compound. This measurement is crucial for high-precision mass spectrometry applications where accurate mass determination is needed. This could indicate a problem with the mass spectrometer's calibration, tuning, or performance.

  • Line #752: The anticancer effects of all these components were demonstrated. (reference needed)
  • There is repetition in lines #170, 171.
  • Ether extract has significant effects on various biomarkers, but these actions were not correlated to their active constituents. The reasons for the pronounced effects for one kind of biomarker by alcohol extract or by ether extract need to be given.
  • Aerial parts of plants, such as leaves, flowers, and stems, generally contain a higher concentration and diversity of secondary metabolites compared to seeds. In a study, 42 compounds were identified in chia aerial parts (https://www.mdpi.com/2223-7747/12/5/1062), while in the present study, 121 and 18. Kindly give reasons how your study is different.

This manuscript contains significant findings but requires substantial revisions for clarity, accuracy, and completeness. Addressing the comments and recommendations provided will enhance the manuscript's quality and suitability for publication.

Comments on the Quality of English Language

Minor correction (grammar) is rquired 

Author Response

Author’s Response for Reviewer 2

Responses to reviewers’ comments for “The therapeutic impact of chia seed (Salvia hispanica) extracts on tumor growth regression in 4-(Methylnitrosamino)-1-(3-pyridyl)-1-butanone (NNK)- induced lung cancer in rats” (Manuscript ID: pharmaceuticals-3127019.).

Dear Editor,

Firstly, I would like to extend my profound thanks to the reviewer for reading and reviewing the manuscript with great care and depth. We also thank you for your comments, which will enable us to refine and improve our manuscript. We hope that you will accept it after revision. According to the reviewers' suggestions, the manuscript has been revised carefully. The detailed corrections raised by the reviewers have been addressed point by point and highlighted in green in the revised manuscript.

Comments and Suggestions for Authors

Dear Authors

The following points need to be addressed. 

Commented 1: Title: The therapeutic word is not suitable in the title as establishing a molecule or formula for therapeutic purposes requires rigorous studies.

Response: We thank the reviewer for pointing this out.

  • We changed the title to “Chia seed (Salvia hispanica) Attenuates Chemically-Induced Lung Carcinomas in Rats Through Suppression of Proliferation and Angiogenesis”
  • We hope the title is more suitable.

Commented 2: Abstract: Add a conclusive statement or outcome of the study.

Response: We are sorry.

  • Limiting the abstract to a small number of words (200 words) did not enable us to provide more data about our study. However, we were able to give a brief conclusion of the study’s outcomes: “Finally, we can conclude that chia seeds have an anti-lung cancer effect with a good safety margin” (page 1, lines 35-36).
  • We hope this addition is more suitable.

Commented 3: Introduction: Add recent statistics on the incidence and mortality rates of lung cancer globally to highlight the significance of the disease.

Response: We added new information in 2022.

  • “Lung cancer was the most frequently diagnosed cancer in 2022, responsible for almost 2.5 million new cases, or one in eight cancers worldwide (12.4% of all cancers globally), followed by cancers of the female breast (11.6%). Lung cancer was also the leading cause of cancer death, with an estimated 1.8 million deaths (18.7%), followed by colorectal (9.3%) and liver (7.8%) cancers.” Page 2, lines 57-61.

Ref: on page 45, lines 1146- 1148.

Commented 4: Results: There is no need for the first paragraph in the results section; it is already given in the methods section.

  • Weight and lung histopathology are not biomarkers. Correct this.

Response: The comment was done.

  • The statement was corrected to “To evaluate the anti-lung cancer effect, we estimated three anti-lung cancer biomarkers: Intercellular Adhesion Molecule-1(ICAM-1 and c-MYC and MMP9 genes.Page 4, lines 126-127.
  • The second paragraph seems to be the summary of the work, which is not required in the results section.

Response:

  • In this paragraph, we have summarized the information to be explained in this section to facilitate its reading. However, we will delete it if necessary.

Commented 5:  Table 2: Improve the caption to clearly describe the table content and the meaning of superscripts. For example: "Values with the same superscript letter within a column are not significantly different (P ≤ 0.05)."

Response:

  • The comment was done.
  • The table title was changed to “Table. 2. The Effect of Chia ether and ethanolic extracts on liver functions of NNK-induced lung cancer in rats during five months.”
  • Also, 3. The Effect of Chia ether and ethanolic extracts on renal functions of NNK-induced lung cancer in rats during five months.” Page 9.

Commented 6:  Section 2.2.8: Histology is primarily used to diagnose and characterize lung cancer, not as a biomarker per se. Please correct this.

Response:

  • The comment was made, and the statement was deleted. Page 10.

Commented 7: Section 2.3: Herbal medicines and derivative products are widely employed as therapeutics in many nations. Please revise as "Herbal medicines and their derivative products are widely used as therapeutic agents in many countries."

Response: We are so sorry.

  • We changed and revised section 2.3: to "Herbal medicines and their derivative products are widely used as therapeutic agents in many countries."

Commented 8:  The statement, "Various mass spectrometric techniques have gradually been used to investigate medicinal plants and profile their secondary metabolites. Chromatography has recently embraced high-performance liquid chromatography/quadrupole time-of-flight mass spectrometry (HPLC/QTOF-MS) technology. Its excellent resolution, rapidity, and sensitivity make it an ideal separation technique" is mostly accurate but can be refined for clarity and precision. Delete this due to repetition:

Response: We are so sorry.

  • We deleted the repetition

Commented 9:  “The initial motivation for this work is to use Salvia hispanica (Chia) seeds belonging to the mint family (Lamiaceae) to identify a viable cure for a dangerous lung cancer disease. Among our other objectives in this study is to investigate the chemical profile of the most active extracts.”This is not the right claim for the study: “Therefore, our goal was to identify the most effective anti-cancer lung compounds.” Rectify this.

Response:Many thanks and appreciation to the reviewer.

  • We Rectified this (( The initial motivation for this work is to use Salvia hispanica (Chia) seeds belonging to the mint family (Lamiaceae) to identify a viable cure for a dangerous lung cancer disease. Among our other objectives in this study is to investigate the chemical profile of the most active extracts.)) to The initial motivation for this work is to use Salvia hispanica (Chia) seeds belonging to the mint family (Lamiaceae) to identify a viable cure for a dangerous lung cancer disease. Therefore, our goal was to identify the most effective anti-cancer lung compounds.
  • We hope we have done the suitable correction

Commented 10: Font size throughout the manuscript needs to be uniform.

Response: The comment was done.

Commented 11:  Check Table 4 and correct; I think it is diterpene, not “Diterbene.”

Response: We thank the reviewer for pointing this out. We Checked Table 4 and corrected Diterbene to  diterpene

Commented 12: Alkaloid Section: The alkaloid section needs to be revisited. The abstract and discussion inform that one alkaloid is present, but in the alkaloid section, the number mentioned is 04, and the names assigned are 03. Additionally, it should be noted that adenosine is not an alkaloid. The claim that chia seeds contain alkaloids would require substantial scientific evidence from phytochemical analysis studies. If such evidence is not available, the assumption should be that chia seeds do not contain notable levels of alkaloids.

Response 1:  We thank the reviewer for pointing this out. We removed the adenosine compound from Table (4 ) and the text of discussion section for alkaloids. We  added the statment (chia seeds do not contain notable levels of alkaloids.) at the beginning of the alkaloid section.

Commented 13: What reference compounds were used in the LCMS analysis?

Response:

  • The reference was [18,19] on page 43, line 1055.
  1. Abd Elkarim, A.S.; Ahmed, A.H.; Taie, H.A.A.; Elgamal, A.M.; Abu-elghait, M.; Shabana, S. Synadenium grantii hook f.: HPLC/QTOF-MS/MS tentative identification of the phytoconstituents, antioxidant, antimicrobial and antibiofilm evaluation of the aerial parts. J. Chem. 14,2021B, 811-828
  2. El-Newary, S.A.; Abd Elkarim, A.S.; Abdelwahed, N.A.; Omer, E.A.; Elgamal, A.M.; Elsayed, W.M. Chenopodium murale Juice shows anti-Fungal efficacy in experimental oral candidiasis in immunosuppressed rats in relation to its chemical profile. Molecules2023, 28,4304.

Page 46, lines 1188-1193

Commented 14:  Figure 6: The need for optimization to achieve better resolution and accurate identification of compounds. 1.136 - 1.322 minutes: Early peaks showing potential overlap. 13.435 - 15.324 minutes: Several closely eluting peaks in the middle region. 19.750 - 20.209 minutes: Late peaks with close elution times. The provided chromatogram may not be acceptable for publication in its current state due to insufficient resolution of several peaks. The lack of clear separation between closely eluting compounds can compromise the specificity, accuracy, and quantitative reliability of the analysis. To meet the publication standards, the chromatographic conditions should be optimized to achieve better separation and resolution of the peaks. This could involve adjusting the gradient program, changing the column, altering the flow rate, or modifying the mobile phase composition.

Response: We thank you We have learned a lot from your comments, which have improved the manuscript and made it better.

  • We added the better resolution of chromatograms in figure 6.
  • We hope we have done the suitable correction

Commented 15: Figure 7: The separation in the positive ionization mode is not fully acceptable for publication due to insufficient resolution of several peaks.

Response:

  • We deleted the positive mode chromatogram in Figure 7.
  • We hope we have done the suitable correction

Commented 16: Compound Classification: L-Arginine is not a fatty acid. It is an amino acid.

  • Suberic acid, also known as octanedioic acid, is a dicarboxylic acid. It is not a fatty acid.
  • Citraconic acid, also known as methylmaleic acid, is an unsaturated dicarboxylic acid. It is not a fatty acid. It will be better to recheck the classification
  • The grouping of certain compounds needs to be revised or grouped under miscellaneous if they do not fit a particular class of compounds.

Response: We thank the reviewer for pointing this out and we so sorry.

  • L-Arginine corrected and classified as amino acid in table (5) and text.
  • Suberic acid, also corrected and classified as dicarboxylic acid in table (5) and text
  • Citraconic acid, also corrected and classified as an unsaturated dicarboxylic acid

Commented 17: Diff (ppm) Values: In Table 4 & 5, Diff (ppm) values for up to 7 peaks are higher than 5 ppm, indicating poor precision. A value for anthocyanins is 42. A deviation of 42 ppm indicates that the measured m/z value is significantly different from the expected value, which could lead to incorrect identification of the compound. This measurement is crucial for high-precision mass spectrometry applications where accurate mass determination is needed. This could indicate a problem with the mass spectrometer's calibration, tuning, or performance.

Response:  We so sorry.

  • We rechecked and the Diff (ppm) values for anthocyanins and all compounds listed in table 4. It was 4.24 not 42 for Delphinidin-3-O- glucopyranoside and 65 for Delphinidin-3-O-(6''-O-alpha-rhamnopyranosyl-beta-glucopyranoside).

Commented 18: Line #752: The anticancer effects of all these components were demonstrated. (reference needed)

Response:

  • We discussed each compound's anticancer effect and added its reference in the next paragraphs (references from 68-81), pages 37-38, lines 800-856.

Commented 19: There is repetition in lines #170, 171.

Response:

  • There is no repetition in the lines. We mean that the two extracts did not significantly affect the c-MYC gene, but alcohol alone significantly affects theMMP9 gene.“On the other hand, as compared to the negative control group, chia alcohol, and ether extracts revealed a nonsignificant decrease in expression levels of the c-MYC gene ( 2 (C), while chia alcohol extract revealed a significant decrease in expression levels of MMP9 gene (P≤ 0.05) (Figure. 2(D).” page 5, lines 169-173.

Commented 20: Ether extract has significant effects on various biomarkers, but these actions were not correlated to their active constituents. The reasons for the pronounced effects of one kind of biomarker by alcohol extract or by ether extract need to be given.

Response:

Dear reviewer:

The effect of the two extracts was similar; sometimes, the ether extract was superior, and at other times, the alcoholic extract was superior. Therefore, we did not compare the extracts. We added a new paragraph about the effect of petroleum ether extract “Also, Chia seed petroleum ether extract exhibited these characteristics due to its chemical composition, which includes considerable quantities of secondary metabolites identified by GC—MS and LC-Mass, such as β-Sitosterol (43.10%) and linolenic acid (67.48%). β-Sitosterol significantly inhibited the growth of A549 cells without harming normal human lung and PBMC cells. Further, BS treatment triggered apoptosis via ROS-mediated mitochondrial dysregulation as evidenced by caspase-3 & 9 activations, Annexin-V/PI positive cells, PARP inactivation, loss of MMP, Bcl-2-Bax ratio alteration, and cytochrome c release. Moreover, generation of ROS species and subsequent DNA strand break were found upon β-Sitosterol treatment. Indeed, β-Sitosterol treatment increased p53 expression in A549 cells. [82] In addition, β-Sitosterol effectively suppressed proliferation and promoted apoptosis of lung cancer using the A549 cell line and A549/anlotinib cell. [83]Linolenic acid exerts significant anticancer effects on multiple cancers, including lung cancer. Its various effects include inhibiting proliferation, inducing apoptosis, suppressing tumor metastasis and angiogenesis, and exerting antioxidant effects. [84]”

Page 38, lines 860-873.

Commented 21: Aerial parts of plants, such as leaves, flowers, and stems, generally contain a higher concentration and diversity of secondary metabolites compared to seeds. In a study, 42 compounds were identified in chia aerial parts (https://www.mdpi.com/2223-7747/12/5/1062), while in the present study, 121 and 18. Kindly give reasons how your study is different.

Response:

Dear reviewer:

 Our study completely differs from the mentioned study. They studied the chemical composition of the aerial part's non-polar fraction. We studied the chemical composition of seeds petroleum ether (using GC-MS and LC-Ms) and ethanolic 70% extracts (using HPLC/ QTF- HR -MS/MS).

This manuscript contains significant findings but requires substantial revisions for clarity, accuracy, and completeness. Addressing the comments and recommendations provided will enhance the manuscript's quality and suitability for publication.

Thank you for your attention and help!

Yours Sincerely,
Samah Ali El-Newary
Associated professor at Aromatic and Medicinal Plants Department, National Research Centre,

El-Tahrir St., Dokki, Giza, 12311, Egypt.

Mobile: 01000464073

Reviewer 3 Report

Comments and Suggestions for Authors

- Adding literature information to the introduction of the abstract may be beneficial for understanding the subject.

- What is the purpose of using the "alcohol, and petroleum ether extracts" item stated in the abstract section as "This study was carried out to study the effects of chia seeds, alcohol, and petroleum ether extracts on lung cancer in vitro and in vivo models"?

- There is confusion of meaning in this sentence. "Cytotoxicity of the chia extracts was studied in lung 22 cancer cell line (A549 cells). In-vitro, Chia alcohol, and ether extracts exhibited a strong cytotoxic effect as a low IC50, 16.08, and 14.8 µg/ml, respectively, in comparison to Dox (IC50, 13.6µg/ml) against A549 cells after 48 hours." should be corrected.

-The clear names of abbreviations should be added.

"Chia seeds fought lung cancer via suppression of proliferation, angiogenesis, inflammation, and activation apoptosis." The subparameters expressed in this sentence should be expressed in parentheses. For example, inflammation (such as IL-6).

Although the authors used this statement ". Despite the high nutritional value of chia seeds, few studies have addressed their therapeutic functions. From this standpoint, the current study was conducted." it is seen that there are many studies in the literature (https://scholar.google.com/scholar?hl=tr&as_sdt=0%2C5&q=%22chia+seeds%22+%22in+vivo%22+%22in+vitro%22&btnG=). Therefore, the authors should explain the novelty of this study better.

-Why were only 48-hour experiments set up in in vitro studies and not 24, 72 and 96-hour ones?

-"In vivo study on NNK-induced lung cancer in rat model" Let's either remove "in vivo" or "in rat model" from this sentence.

-Why were 20 rats used in in vivo studies? While the general acceptance is 7-8 rats.

-In in vivo studies, how were doses determined.

-The article's spelling is bad and should be seriously reviewed.

-Why were in vitro and in vivo studies performed together in this study?

- It is said that the applied chia ethanolic extract was applied orally. It is not possible to say that a substance given orally is effective in rats with lung cancer. Can an inhalable form of this substance be produced?

- The discussion section is very messy and lacks coherence. It should be rewritten.

Comments on the Quality of English Language

Low.

Author Response

Author’s Response for Reviewer 3

Responses to reviewers’ comments for “The therapeutic impact of chia seed (Salvia hispanica) extracts on tumor growth regression in 4-(Methylnitrosamino)-1-(3-pyridyl)-1-butanone (NNK)- induced lung cancer in rats” (Manuscript ID: pharmaceuticals-3127019.).

Dear Editor,

Firstly, I would like to extend my profound thanks to the reviewer for reading and reviewing the manuscript with great care and depth. We also thank you for your comments, which will enable us to refine and improve our manuscript.We hope that you will accept it after revision. According to the reviewers' suggestions, the manuscript has been revised carefully. The detailed corrections raised by the reviewers have been addressed point by point and highlighted in blue in the revised manuscript.

Comments and Suggestions for Authors

Commented 1: Adding literature information to the introduction of the abstract may be beneficial for understanding the subject.

Response: We are sorry.

  • Limiting the abstract to a small number of words (200 words) did not enable us to provide more data about our study. However, we added a short statement as “Lung cancer is the most diagnosed and deadliest cancer worldwide, with 2.2 million cases in 2020 resulting in 1.8 million deaths. Those statistics motivated us to introduce a new natural product feasible in lung cancer therapies.”.Page 1, lines 21-23.

Commented 2:  What is the purpose of using the "alcohol, and petroleum ether extracts" item stated in the abstract section as "This study was carried out to study the effects of chia seeds, alcohol, and petroleum ether extracts on lung cancer in vitro and in vivo models"?

Response: We thank the reviewer for pointing this out.

  • The statement was poorly written.
  • The statement was rewritten to be “This study was carried out to study the effects of two extracts from chia seeds(70% ethanol and petroleum ether) on lung cancer in vitro and in vivo models”Page 1, lines 24-2

Commented 3: There is confusion about the meaning of this sentence. "Cytotoxicity of the chia extracts was studied in lung 22 cancer cell line (A549 cells). In-vitro, Chia alcohol, and ether extracts exhibited a strong cytotoxic effect as a low IC50, 16.08, and 14.8 µg/ml, respectively, in comparison to Dox (IC50, 13.6µg/ml) against A549 cells after 48 hours." should be corrected.

Response:We thank the reviewer for pointing this out. We corrected this sentence in the text.Page 1, lines 22-27.

Commented 4: The clear names of abbreviations should be added.

Response:We are sorry. Limiting the abstract to a small number of words (200 words) did not enable us to provide more data about our study.

On page 1, lines 30-31, we added the meaning of NNK: “4-(Methylnitrosamino)-1-(3-pyridyl)-1-butanone (NNK)—induced lung cancer.”

All abbreviations were mentioned at the beginning of the results in the second paragraph, lines 132-140.

We hope this modification to be satisfactory.

Commented 5: "Chia seeds fought lung cancer via suppression of proliferation, angiogenesis, inflammation, and activation apoptosis." The subparameters expressed in this sentence should be expressed in parentheses. For example, inflammation (such as IL-6).

Response:We are sorry.

  • Limiting the abstract to a small number of words (200 words) did not enable us to provide more data about our study.
  • We must not exceed the number of words, so we have kept it very brief.

Commented 6: Although the authors used this statement ". Despite the high nutritional value of chia seeds, few studies have addressed their therapeutic functions. From this standpoint, the current study was conducted." it is seen that there are many studies in the literature (https://scholar.google.com/scholar?hl=tr&as_sdt=0%2C5&q=%22chia+seeds%22+%22in+vivo%22+%22in+vitro%22&btnG=). Therefore, the authors should explain the novelty of this study better.

Response:

  • Yes, there are many studies that have investigated the biological activities of chia seeds, as you said, but they are few compared to the fantastic nutritional value of chia seeds. Chia needs more and more studies.
  • Our study is the first to examine the anticancer effect of chia seed extracts, petroleum ether, and 70% ethanol on lung cancer on an in vivo scale. It is also the first study to use this model: 4-(Methylnitrosamino)-1-(3-pyridyl)-1-butanone (NNK) - induced lung cancer in rats. In addition, we studied how shea extracts work in treating lung cancer.
  • Several studies have shown the anti-cancer effect of chia oil in lung cancer, but on an in vitro scale using A549 cell lines.

The statement on pages 4-5, lines 120-125.

  • In this work, all of the separated compounds were detected and tentatively identified for the first time in the ethanol extract of Egyptian Salvia hispanica (Chia) seeds.

The statement on pages 16, lines 401-402.

Commented 7: Why were only 48-hour experiments set up in in vitro studies and not 24, 72 and 96-hour ones?

Response:

  • We treated the chia extracts for 48 hours according to previous studies such as:
  • Ortega AMM and Campos MRS. Effect of Chia Seed Oil (Salvia hispanica L.) on Cell Viability in Breast Cancer Cell MCF-7. Proceedings 2020, 53, 18; doi:10.3390/proceedings2020053018
  • kumar DG, Perumal PC, Kumar K, Muthusami S, Gopalakrishnan VK. Dietary Evaluation, Antioxidant and Cytotoxic Activity of Crude Extract from Chia Seeds (Salvia hispanica L.) against Human Prostate Cancer Cell Line (PC-3). International Journal of Pharmacognosy and Phytochemical Research 2016; 8(8); 1358-1362.

Commented 8: "In vivo study on NNK-induced lung cancer in rat model" Let's either remove "in vivo" or "in rat model" from this sentence.

Response:

  • The comment is done. We removed the rats.

Commented 9: Why were 20 rats used in in vivo studies? While the general acceptance is 7-8 rats.

Response:

  • This study was extended for 40 weeks (a very long period), which prompted us to work on a large sample to avoid the natural loss of animals due to the long period.

Commented 10: In in vivo studies, how were doses determined.

Response:

  • The tested dose (500 mg/kg) was selected as a tenth of LD50.
  • We determined the extracts' LD50, which was 5 g/kg. The data are on page 42, lines 967-980, in the section Materials and Methods.

Commented 11: The article's spelling is bad and should be seriously reviewed.

Response:We thank the reviewer for pointing this out.

  • The manuscript was seriously reviewed.

Commented 12: Why were in vitro and in vivo studies performed together in this study?

Response:

  • We performed the in vitro cytotoxicity screening to assess the activity of these extracts before testing them in the in vivo study.

Commented 13:  It is said that the applied chia ethanolic extract was applied orally. It is not possible to say that a substance given orally is effective in rats with lung cancer. Can an inhalable form of this substance be produced?

Response:

Dear reviewer:

In the current study, we applied chia extracts orally, and they were effective against lung cancer. In our future studies, we will take this into consideration.

Commented 14: The discussion section is very messy and lacks coherence. It should be rewritten.

Response:

Thank you for your attention and help!

Yours Sincerely,
Samah Ali El-Newary
Associated professor at Aromatic and Medicinal Plants Department, National Research Centre,

El-Tahrir St., Dokki, Giza, 12311, Egypt.

Mobile: 01000464073

Round 2

Reviewer 1 Report

Comments and Suggestions for Authors

The author made great modifications to the manuscript to meet the reviewer's requirements. Thus, it should be accepted for publication now.

Reviewer 2 Report

Comments and Suggestions for Authors

The authors have addressed all the suggested revisions, and the manuscript may be considered for publication now.

Reviewer 3 Report

Comments and Suggestions for Authors

Suggested corrections have been made.